# TAIL: Task-specific Adapters for Imitation Learning with Large Pretrained Models

**Zuxin Liu**[1], **Jesse Zhang**[2], **Kavosh Asadi**[3], **Yao Liu**[3], **Ding Zhao**[1], **Shoham Sabach**[3], **Rasool Fakoor**[3]

[1]Carnegie Mellon University, [2] University of Southern California, [3] Amazon Web Services

## Abstract

The full potential of large pretrained models remains largely untapped in control domains like robotics. This is mainly due to data scarcity and computational challenges associated with training or fine-tuning large models for such applications. Prior work mainly emphasizes either effective *pretraining* of large models for decision-making or single-task adaptation. But real-world problems will require data-efficient, *continual adaptation* for new control tasks. Recognizing these constraints, we introduce TAIL (**T**ask-specific **A**dapters for **I**mitation **L**earning), a framework for efficient adaptation to a stream of new control tasks. Inspired by recent advancements in parameter-efficient fine-tuning in language domains, we explore efficient fine-tuning techniques—e.g., Bottleneck Adapters, P-Tuning, and Low-Rank Adaptation (LoRA)—in TAIL to adapt large pretrained models for new tasks with limited demonstration data. Our extensive experiments comparing prevalent parameter-efficient fine-tuning techniques and adaptation baselines suggest that TAIL with LoRA can achieve the best post-adaptation performance with only 1% of the trainable parameters of full fine-tuning while avoiding catastrophic forgetting and preserving adaptation plasticity in continual learning settings.

## 1 Introduction

A desired property of an autonomous agent is the ability to adapt efficiently to novel tasks. In vision and language domains, large pretrained models have demonstrated adaptation to new tasks with just a few examples through prior knowledge obtained from internet-scale datasets (Brown et al., 2020; Radford et al., 2021; Touvron et al., 2023). Similar methods have also been applied in decision-making and control applications (Brohan et al., 2022; Driess et al., 2023; Brohan et al., 2023). However, new control tasks are more difficult to adapt to than the aforementioned vision and language domains due to (1) the lack of internet-scale control data and (2) how optimal actions can vary significantly from task-to-task, even under shared observation spaces. As such, these large-scale decision-making models still rely on a close alignment between training and testing tasks.

In contrast, agents deployed in challenging environments need to adapt to major task variations—take, for example, a general household robot. Equipped with a factory-pretrained policy, the robot will be employed in unique ways by every household. Thus, the robot will need to *continually adapt* in order to best serve each one, e.g., by fine-tuning its capabilities on a few demonstrations (Chebotar et al., 2021; Lu et al., 2021; Kalashnkov et al., 2021; Chen et al., 2021a; Yao et al., 2024). Because most prior decision-making papers adapt to new tasks by fine-tuning the entire model (Gupta et al., 2022; Bousmalis et al., 2023; Zhang et al., 2023a;b; Collaboration et al., 2023; Liu et al., 2023c), mastering each new skill requires great computational cost and often leads to catastrophic forgetting of old ones. An alternative approach would be to store a separate policy per new task, which leads to unreasonable storage requirements. Some prior work investigates efficient adaptation of large models to a single task suite (Liang et al., 2022; Schmied et al., 2023; Sharma et al., 2023), but this realistic continual learning setting brings out additional problems to consider, warranting further investigation. What would be the best way for agents to *efficiently adapt* to a stream of novel tasks without having to trade off computation, storage, and performance on older tasks?

To answer this question, we propose **T**ask-specific **A**dapters for **I**mitation **L**earning, shown in Fig. 1, a framework for efficient adaptation to new control tasks. Through TAIL we (1) effectively incorporate lightweight adapter modules into pretrained decision-making models and (2) comprehensively com-

pare efficient adaptation techniques implemented in TAIL in a continual imitation learning setting. Notably, we examine parameter-efficient adaptation techniques (PEFT) used for large language models; we explore the potential of adapters (Houlsby et al., 2019), prefix tuning (Li & Liang, 2021), and low-rank adaptation (LoRA) (Hu et al., 2021) in fostering efficient and continual adaptation in large pretrained decision-making models. These works stand out as they introduce a small number of *new* parameters which help: avoid catastrophic forgetting, maintain training plasticity for continual learning, avoid overfitting with limited adaptation data, and reduce computational and memory burden. Investigating these works in control tasks for a realistic continual learning setup specifically is important because, unlike in language domains, test task losses are often not proportional to test task performance (Ross et al., 2011; Ke et al., 2020)—efficient adaptation insights from language models may not transfer to decision-making ones. Thus, independent investigation of these adaptation techniques for decision-making is crucial for deploying continually adapting agents in the real world.

We compare PEFT techniques implemented in TAIL against commonly used adaptation methods in the imitation learning literature. In our experiments, we discover that TAIL with LoRA leads to the best post-adaptation performance as it preserves the original pretrained representations while being resilient against overfitting in the limited-data regime. These capabilities are especially important for agents operating in new, challenging environments, such as the aforementioned household robots. Our analysis also reveals important insights into the strengths and limitations of each adaptation strategy. Instead of performing full fine-tuning of the entire model, TAIL only introduces a small number of additional parameters without making changes to the original model. These additional parameters make up a mere **1.17**% of the size of the original model. Importantly, this results in approximately **23**% less GPU memory consumption to achieve **22**% higher forward adaptation success rate than full fine-tuning while avoiding catastrophic forgetting. Notably, these results are contrary to many results from the vision and language model literature which show that full fine-tuning works better (He et al., 2022; Mao et al., 2022; Chen et al., 2022; Schmied et al., 2023).

In summary, this work bridges a crucial gap in research into efficient and continual adaptation for pretrained decision models by introducing a framework for continual imitation learning, TAIL, and thoroughly analyzing the effects of different efficient adaptation methods. Comprehensive experiments demonstrate that TAIL outperforms standard continual learning and prior single-task adaptation baselines.

## 2 RELATED WORK

**Pretrained Models for Control.** Researchers have long studied the use of pretrained models for better downstream transfer to related tasks (Bozinovski & Fulgosi, 1976; Schmidhuber, 1992; Dietterich et al., 1997). Recent works have examined using the representations learned by pretrained visual models for control (Shridhar et al., 2022; Nair et al., 2022; Ma et al., 2022; 2023; Majumdar et al., 2023a). These methods leverage representations acquired from large task-agnostic datasets, such as Ego4D (Grauman et al., 2022), or through self-supervised objectives. However, there's evidence that simply utilizing these pretrained features may not be as useful for downstream task performance (Hansen et al., 2022). Meanwhile, another recent line of work directly trains large pretrained models for control (Brohan et al., 2022; Reed et al., 2022; Driess et al., 2023; Jiang et al., 2023; Brohan et al., 2023; Bousmalis et al., 2023). These methods either do not attempt adaptation to new tasks, or perform expensive full-fine-tuning for adaptation. In contrast, our method, TAIL, is a framework for efficient adaptation of decision-making models, like the aforementioned large pretrained control models, and investigates ways to adapt such models efficiently to multiple new tasks.

**Parameter-Efficient Fine-Tuning (PEFT).** PEFT has gained traction as a way to adapt pretrained models without significantly increasing parameters. Rebuffi et al. (2018) demonstrated that residual adapters for smaller, CNN-based vision models are effective in non-control supervised learning settings. More recently, transformer-focused techniques such as transformer adapter modules (Houlsby et al., 2019), LoRA (Hu et al., 2021), and prompt tuning (Li & Liang, 2021) incorporate lightweight modules or prompts optimized for downstream tasks, all while preserving the original model weights. PEFT offers several advantages over full fine-tuning: it's faster, less susceptible to overfitting, retains prior capabilities, and facilitates efficient task-switching. While PEFT has been successful in both language and vision domains (Chen et al., 2022; Schmied et al., 2023), its continuous adaptation for large decision-making models is not yet thoroughly examined. Liang et al. (2022); Sharma et al. (2023), Xu et al. (2022), and Xu et al. (2023) propose the use of adapters, prompt-tuning, and hyper-network in robotics settings, but they do not examine other PEFT methods and focus on

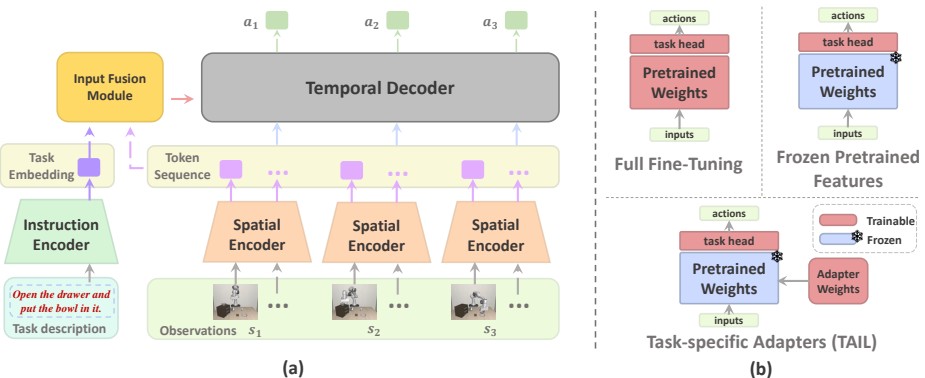

Figure 1: **(a)**: The multi-modal, transformer policy architecture we utilize for pretraining. We encode language task descriptions with a pretrained CLIP instruction encoder and image observations with a pretrained CLIP spatial encoder. We additionally encode state observations (not pictured) which, along with the observation embeddings, are embedded into a sequence of tokens used by the temporal decoder transformer to predict single-step action distributions. We include an input fusion module to explicitly combine the task embedding with the observation token sequence for better instruction-following ability. **(b)**: The three types of fine-tuning paradigms we test, with TAIL at the bottom right. For further architecture details, see Appendix Sec. A.

adaptation to a single task suite. We instead examine the performance of various state-of-the-art PEFT techniques implemented with TAIL in the *continual learning* scenario.

**Continual Learning.** Continual learning in control (Thrun & Mitchell, 1995; McCloskey & Cohen, 1989; Fu et al., 2022) is a long-studied problem with applications to many real-world situations. In general, agents should be able to transfer knowledge (e.g., by continually fine-tuning) or experience (e.g., training data) from previously learned tasks to new tasks (Lopez-Paz & Ranzato, 2017; Traoré et al., 2019; Fakoor et al., 2020; Caccia et al., 2023). However, with large pretrained models trained on large datasets, fine-tuning the entire model is computationally costly yet risks catastrophic forgetting, and transferring training data from other tasks is too memory inefficient in the face of a large stream of new tasks. Therefore, we present a study into efficient fine-tuning techniques which, when integrated with TAIL, can help inform future research of continual learning.

## 3 PRELIMINARIES

In this section, we introduce our problem setting (Sec. 3.1), review large, pretrained models for decision-making (Sec. 3.2), and discuss traditional adaptation methods in this area (Sec. 3.3).

### 3.1 CONTINUAL IMITATION LEARNING

The agent encounters a sequence of $K$ tasks, denoted as $\{\mathcal{T}_1, \ldots, \mathcal{T}_K\}$. Each task $\mathcal{T}_k = (\mu_k^0, g_k)$ is characterized by an initial state distribution $\mu_k^0$ and a goal predicate $g_k$. Goals for tasks can be specified using language instructions, providing clear context (Jang et al., 2021; Zhang et al., 2023a). For every task $\mathcal{T}_k$, the agent receives $N$ demonstration trajectories $\mathcal{D}_k = \{\tau_k^1, \ldots, \tau_k^N\}$. In this paper, we use the standard behavioral cloning loss to optimize the agent's policy $\pi$ over these demonstrations, however we note that TAIL can be used with other training objectives as well:

$$\hat{\boldsymbol{\theta}} = \min_{\boldsymbol{\theta}} \sum_{k=1}^{K} \mathbb{E}_{s_t, a_t \sim \mathcal{D}_k} \left[ \sum_{t=0}^{l_k} \mathcal{L} \left( \pi(a|s_{\leq t}, \mathcal{T}_k; \boldsymbol{\theta}), a_k^t \right) \right]. \tag{1}$$

Here, $\mathcal{L}$ is a supervised action prediction (e.g., mean squared error or negative log likelihood) loss, $l_k$ is the length of demonstrations for task $\mathcal{T}_k$, and $\boldsymbol{\theta}$ refers to the *learnable parameters* of the network. Notably, after learning task $\mathcal{T}_k$, the agent cannot access *additional* data from preceding tasks. This presents a continual learning challenge, emphasizing the importance of transferring knowledge across tasks without the risk of catastrophic forgetting (McCloskey & Cohen, 1989).

### 3.2 PRETRAINED DECISION-MAKING MODELS

Here, we briefly describe common features of large pretrained decision-making model architectures used for embodied agents. We incorporate key components shared amongst these models into the architecture of the model that we pretrain to evaluate efficient adaptation, pictured in Fig. 1(a).

**Transformer Backbone.** Most recent work training large-scale decision-making models (Brohan et al., 2022; Shafiullah et al., 2022; Brohan et al., 2023) utilize a transformer backbone (Vaswani et al., 2017) that attends to tokenized observations from prior timesteps. We adopt a standard GPT-2 (Radford et al., 2019a) transformer decoder (Fig. 1(a), temporal decoder) with separate encoders for each input modality and continuous action distribution outputs.

**Pretrained Input Encoders.** Encoders pretrained on large, diverse datasets can produce rich, well-structured embeddings which make it easier to learn the downstream tasks (Jang et al., 2021; Brohan et al., 2022). Therefore, we utilize pretrained CLIP image and textual encoders (Radford et al., 2021).

**Input Modality Fusion.** The idea of explicitly "fusing" different input modalities has seen great success not only in domains like vision and language (Perez et al., 2017), but also in agent learning (Jang et al., 2021; Brohan et al., 2022). Similarly, we utilize FiLM layers (Perez et al., 2017) (Fig. 1(a), input fusion module) to fuse language task specifications with observations.

### 3.3 ADAPTING PRETRAINED MODELS FOR NEW TASKS

One standard adaptation method in prior research is full fine-tuning (FFT) of all model parameters (Fig 1(b), top left). Though straightforward, it is resource-intensive and prone to overfitting with limited data (Bousmalis et al., 2023). There is also a risk of distorting pretrained features, resulting in the loss of prior tasks—a phenomenon known as **catastrophic forgetting** (McCloskey & Cohen, 1989). Evidence also suggests that extensive fine-tuning might undermine a model's rapid adaptability to new tasks, an effect referred to as the loss of **model plasticity and capacity** (Kumar et al., 2022; Lyle et al., 2022; Kumar et al., 2023). Such issues become more prominent in continual learning contexts (Lopez-Paz & Ranzato, 2017). Moreover, duplicating a sizable model for each subsequent task is neither efficient nor practical due to storage limitations.

Another standard adaptation method is the use of frozen pretrained features (FPF, Fig 1(b) top right). FPF ensures the retention of knowledge acquired from previous tasks by tuning a task-specific head. However, as noted in Sharma et al. (2023), it is not expressive enough for out-of-distribution or especially complex tasks. Given these challenges, there's a clear need for a more advanced fine-tuning paradigm that addresses catastrophic forgetting while maintaining model plasticity for adapting to new tasks, all in a data and computationally resource-efficient manner.

## 4 TASK-SPECIFIC ADAPTERS FOR IMITATION LEARNING

In this section, we outline how we perform efficient adaptation on pretrained models through our **T**ask-specific **A**dapters for **I**mitation **L**earning framework, depicted in Fig 1(b). Different from the FPF approach which simply substitutes the policy head for every new task, TAIL introduces a small set of new weights, serving as a lightweight plugin to address specific tasks. This concept draws inspiration from parameter-efficient adaptation techniques prevalent in the language model area. These methods offer several advantages as they: (1) add a few parameters (typically between $0.1\% \sim 2\%$) to preserve the original features, thereby enhancing model plasticity for continual learning and avoiding catastrophic forgetting (Kumar et al., 2023), (2) are resilient to overfitting when adaptation data is scarce, (3) are more computationally and storage-efficient than FFT.

Next, we delve into three prominent weight integration techniques for Transformer-based pretrained models in Sec. 4.1, followed by a case study illustrating the application of this framework in continual imitation learning scenarios in Sec. 4.2.

### 4.1 ADAPTER WEIGHTS INTEGRATION

The concept of an adapter can be best conceptualized as a modular plugin to the base model, customized for specific downstream tasks, that does not affect the model's pretrained representations. We mainly explore three prevalent styles of integration for TAIL: **Parallel** (Hu et al., 2021), **Sequential** (Houlsby et al., 2019; Sharma et al., 2023), and **Prefix Token** (Li & Liang, 2021; Lester et al., 2021; Liu et al., 2023b), all of which are showcased with a Transformer block in Fig. 2. Parallel and sequential integration techniques are generally applicable to any model with feedforward layers, while the prefix token style method is especially tailored for Transformers.

Given a pretrained model, let's consider *one* layer weight matrix in it, denoted as $\boldsymbol{W} \in \mathbb{R}^{d \times k}$. Its input and output hidden states are $h_{in} \in \mathbb{R}^d$ and $h_{out} \in \mathbb{R}^k$, respectively. We have $h_{out} = \boldsymbol{W}^\top h_{in}$. Next, we detail how to apply parallel and sequential insertions to the pretrained weight matrix.

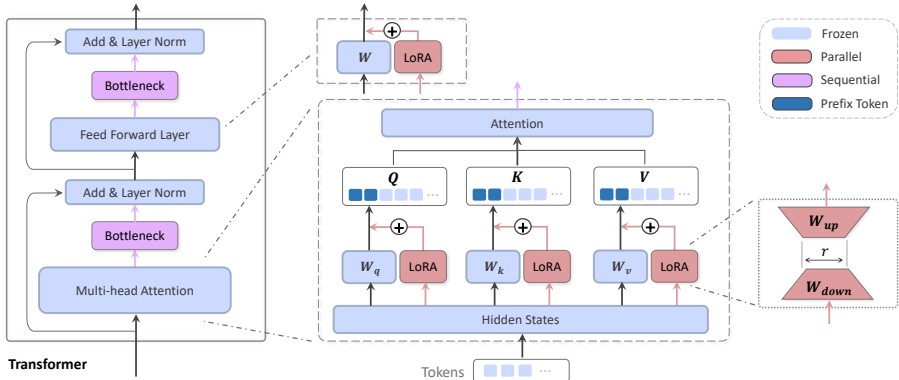

Figure 2: Demonstration of three weight integration styles of TAIL for a Transformer block: sequential (bottleneck adapter), parallel (LoRA), and prefix token (prefix/prompt-tuning).

**Parallel Integration (LoRA).** This integration method, often associated with Low-Rank Adaptation (LoRA) (Hu et al., 2021), introduces trainable low-rank matrices $\boldsymbol{W}_{down} \in \mathbb{R}^{d \times r}$ and $\boldsymbol{W}_{up} \in \mathbb{R}^{r \times k}$. Here, $r \ll \min(d, k)$ represents the rank and is usually much smaller than the dimensions of the original matrix. These matrices are typically integrated in parallel with the original weight matrix $\boldsymbol{W}$ through addition, as shown as LoRA in Fig. 2:

$$h_{out} = \boldsymbol{W}^\top h_{in} + \alpha \boldsymbol{W}_{up}^\top \boldsymbol{W}_{down}^\top h_{in}, \tag{2}$$

with $\alpha$ being a hyperparameter to modulate task-specific adjustments. The above equation can also be formulated as: $h_{out} = (\boldsymbol{W} + \alpha \boldsymbol{W}_{down} \boldsymbol{W}_{up})^\top h_{in} = (\boldsymbol{W} + \alpha \Delta \boldsymbol{W})^\top h_{in}$, where $\Delta \boldsymbol{W}$ denotes the weight modifications for new tasks, and thus the columns of $\boldsymbol{W}_{down}$ and $\boldsymbol{W}_{up}$ can be interpreted as a new basis that contains task-specific knowledge. As observed by Aghajanyan et al. (2020), despite projecting to a condensed subspace with small "intrinsic dimensions," pretrained models can still learn effectively. By introducing the two low-rank matrices, the original weight matrices $\boldsymbol{W}$ can be adeptly tailored with a minimal increase in parameters. Though LoRA was originally crafted for large language models—specifically for the query and value projections matrices $W_Q$ and $W_V$ in multi-head attention (Hu et al., 2021)—it is easily applied to other linear layers as well, such as the Transformer's feedforward layers (Chen et al., 2022).

**Sequential Integration (Bottleneck Adapter).** Renowned in the language model domain, the Bottleneck Adapter introduces bottleneck layers within the model (Houlsby et al., 2019; Sharma et al., 2023) by appending a trainable bottleneck layer after the feedforward network in each Transformer layer. Similar to LoRA, this bottleneck consists of down and up projections, $\boldsymbol{W}_{down}$ and $\boldsymbol{W}_{up}$, which first shrink then restore the dimensions of token hidden states. Formally, for the feedforward network's input $h_{in}$ and a bottleneck size $r$, the output $h_{out}$ is:

$$h_{out} = \boldsymbol{W}_{up}^\top \phi \left( \boldsymbol{W}_{down}^\top (\boldsymbol{W}^\top h_{in}) \right), \tag{3}$$

where $\phi$ denotes a nonlinear activation function. The Bottleneck Adapter (Fig. 2) acts as a filter, isolating relevant information for specific tasks. Yet, filtering often requires a larger bottleneck size compared to that of LoRA, leading to more parameters. Additionally, the sequential insertion can increase latency compared to the parallel nature of LoRA (Hu et al., 2021).

**Prefix Token Integration (Prefix & Prompt-Tuning).** In this style, a set of learnable prefix tokens are appended or prepended to the input sequence (Li & Liang, 2021; Lester et al., 2021; Liu et al., 2023b). Let's consider an input sequence $\mathbf{s} \in \mathbb{R}^{n \times d}$, where $n$ is the sequence length and $d$ is the embedding dimension. The prefix tokens can be represented as $\mathbf{p} \in \mathbb{R}^{m \times d}$, where $m$ denotes the number of prefix tokens. These vectors act like virtual tokens which the original tokens can attend to. They are initialized and learned during the task-specific adaptation phase. The modified input sequence, after appending the prefix tokens, can be expressed as $\mathbf{S} = [\mathbf{p}; \mathbf{s}] \in \mathbb{R}^{(m+n) \times d}$. The model then processes this extended sequence. These prefix tokens can be viewed as task descriptors that are designed to guide the model towards the desired task-specific behavior (see ■ in Fig. 2).

With adapters, we can treat the optimization from Eq. 1 as one over adapter weights instead, where the model is parametrized by $\hat{\boldsymbol{\theta}} = \{\boldsymbol{\theta}, \boldsymbol{\omega}\}$ and $\boldsymbol{\omega}$ is the set of adapter weights we are optimizing for.

Figure 3: Our task suites for continual imitation learning (excluding LIBERO-10). The robot, placed in a tabletop environment, is equipped with a 6-DOF arm and a parallel gripper. It receives RGB images from two views, joint states, and language instructions, and is tasked with producing continuous actions to control its arm.

### 4.2  TAIL FOR CONTINUAL IMITATION LEARNING

We consider the continual imitation learning problem as a typical application of the proposed TAIL adaptation paradigm. The goal of continual imitation learning is to ensure that the model performs effectively on the current task and without significant degradation of performance in past tasks. Given pretrained model weights, denoted as $\boldsymbol{\theta}$, and a new task $\mathcal{T}_k$ with demonstrations $\mathcal{D}_k = \{\tau_k^1, \ldots, \tau_k^N\}$, we initialize the task-specific adapter weight $\boldsymbol{\omega}_k$ with far less parameters than the base model: $|\boldsymbol{\omega}_k| \lll |\boldsymbol{\theta}|$. The adapter weights are inserted into the model through the integration methods introduced in Sec. 4.1. By optimizing the behavior cloning loss in Eq. 1 w.r.t $\boldsymbol{\omega}_k$ while keeping the pretrained weights frozen, the policy adapts to $\mathcal{T}_k$ without interfering with previous tasks.

To execute a task, the corresponding lightweight adapters are loaded as a plugin of the pretrained network weights. For example, when revisiting a prior task $T_j$, where $j \leq k$, the model is configured to solely activate the $j$-th adapter $\boldsymbol{\omega}_j$. This entire procedure can be streamlined as follows:

1. For an incoming task $\mathcal{T}_k$, acquire the training set $\mathcal{D}_k$, initialize a task-specific adapter $\boldsymbol{\omega}_k$.

2. Combine adapter $\boldsymbol{\omega}_k$ with the base model $\boldsymbol{\theta}$ using either parallel, sequential, or prefix token.

3. Train the adapter on $\mathcal{D}_k$ to optimize Eq. 1 for $\boldsymbol{\omega}_k$, keeping pretrained parameters $\boldsymbol{\theta}$ frozen.

In essence, TAIL ensures task-specific knowledge is contained within the adapters, thereby enabling efficient adaptation without catastrophic forgetting. It's also worth noting that the TAIL framework is flexible. The choice of integration method or the specific architecture of the adapter can be tailored based on the complexity of the task or the available computational resources.

## 5  EXPERIMENTS

In this section, we evaluate TAIL on a wide range of tasks and benchmark its performance against other fine-tuning approaches. We mainly aim to answer the following questions: (1) Which efficient adaptation methods in TAIL work best? (2) Can TAIL prevent catastrophic forgetting of previously learned tasks, while allowing more efficient forward adaptation to new tasks over standard adaptation methods? (3) What are the computational efficiencies gained by using TAIL? Addressing them requires a set of diverse tasks in realistic environments, as we describe in the following section.

### 5.1  DATASETS AND BENCHMARK SUITES

We utilize the LIBERO robotic manipulation continual learning benchmark (Liu et al., 2023a), which features a diverse range of tasks that mirror human daily activities, such as turning on a stove, moving books, and opening drawers. Each task is specified via natural language instructions, for instance, *"Open the top drawer of the cabinet, and put the bowl in it."*

We craft a *pretraining* task suite, named **Kitchen**, involving 40 diverse tasks sourced from the LIBERO-90 dataset's kitchen scenes. We then evaluate *adaptation* to 5 separate task suites. LIBERO contains 3 task suites tailored for continual learning, focusing on evaluating different aspects of knowledge adaptation: the **Spatial** task contains the same objects in each scene but with different spatial layouts; each task in the **Goal** suite has distinct goals (such as open the drawer, or turn on the stove), while keeping the objects and layout fixed; the **Object** suite contains pick-and-place tasks for different objects in the scene but with the same layout. To create a more comprehensive experimental setting, we also create 2 *additional* task suites (from LIBERO-90): **Living Room**, and **Study Room**. We adopt 8 tasks from each of the 5 adaptation task suites, respectively. Finally, we separately evaluate each task sequentially in **LIBERO-10**, a benchmark with 10 challenging long-horizon tasks. See Fig. 3 for task suite examples and Appendix Sec. D for more details.

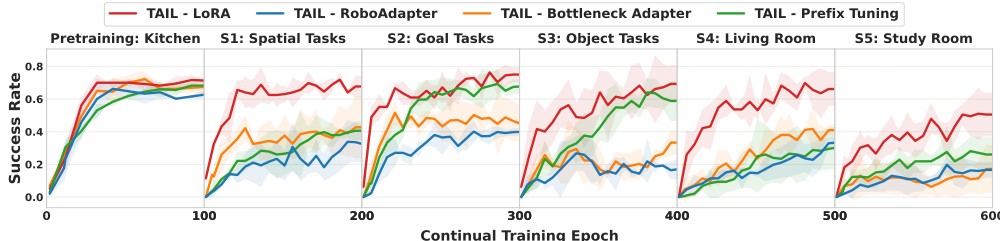

Figure 4: Success rates for different types of adapters under our TAIL framework. None of these methods suffer from catastrophic forgetting, so backward evaluation results are not presented here. LoRA performs best across all tasks, underscoring the benefits of the parallel integration approach.

## 5.2 EXPERIMENT SETUP

**Evaluation metrics.** The primary metric we report is average per-task *suite* success rate, measured by checking if current state aligns with pre-defined goal states. For continual learning, we also assess **Forward Transfer** (FWT) and **Backward Transfer** (BWT) across the curriculum of suites. Following the metric proposed in LIBERO (Liu et al., 2023a), FWT is computed by the maximum success rate one algorithm can achieve when adapting to a new task. We denote FWT at task $k$ as $\mathbf{F}_k$. Meanwhile, BWT measures the success rate increase on previous tasks. Namely, when adapting to the $k$-th task, we record the best FWT model on this task and then evaluate this model on all previous $k-1$ tasks, obtaining success rate $\mathbf{S}_i, 1 \le i \le k-1$. Then we compute the success rate difference between the new model and the best FWT of the previous $k-1$ tasks and then average among them to obtain the BWT metric: $\mathbf{B}_k = \frac{1}{k-1} \sum_{i=1}^{k-1} (\mathbf{S}_i - \mathbf{F}_i)$. For both metrics, higher is better.

**Model architecture.** We adopt the CLIP-base model (Radford et al., 2021) as both the spatial encoder and the language instruction encoder, each with 12 transformer layers. A 6-layer GPT2 structure (Radford et al., 2019b) serves as our temporal encoder, with the FiLM module (Perez et al., 2017) handling input fusion. These components are well-regarded in the literature (Chen et al., 2021b; Brohan et al., 2022; Jiang et al., 2023). Further architectural details can be found in Appendix A.

**Continual Learning Baselines.** We adopt four baselines: Full Fine-Tuning (FFT), Frozen Pretrained Features (FPF) which mirrors the linear probing method (Kumar et al., 2022) but also tunes both the policy head and fusion module *per task*, Experience Replay (ER) (Chaudhry et al., 2019) which uses a 50-50 data split between new and previous task data while adapting to a new task (Rolnick et al., 2019), Elastic Weight Consolidation (EWC) (Kirkpatrick et al., 2017) which regularizes updates of crucial parameters from earlier tasks based on their Fisher information, and PackNet (Mallya & Lazebnik, 2018) which prunes parameters to then be re-learned for every new task. These all use the same model and task conditioning, i.e., language, as TAIL. Further baseline details in Appendix B.1.

**TAIL Adapters.** We use LoRA (Hu et al., 2021), Bottleneck Adapter (Houlsby et al., 2019), and Prefix Tuning (Li & Liang, 2021) to represent parallel, sequential, and prefix integration styles. RoboAdapter (Sharma et al., 2023), a specific implementation for decision-making, stands as another *sequential* integration style. Unlike the Bottleneck Adapter that applies weights at every transformer layer, it introduces weights only at specific transformer layers and exclusively after the feedforward layer. Configuration specifics and more details for these adapters are available in Appendix B.2.

**Training, Adaptation, and Evaluation.** Each task provides 50 successful human demonstrations. These are divided into 40 training trajectories and 10 for validation. *We report success rates over 10 scenes with initial states that are unseen in training*. This limited demonstration setup offers an opportunity to determine which technique is less prone to overfitting in data-restricted conditions. Given our focus on evaluating the adaptation of large pretrained models, we further increase adaptation difficulty by training on and evaluating adaptation performance on all tasks within a task suite simultaneously.[1] We pretrain on **Kitchen** until performance convergence (100 epochs). Subsequent adaptations follow two setups: (1) sequential adaptation across the **Spatial**, **Goal**, **Object**, **Living Room**, and **Study Room** task suites for 100 epochs each, and (2) adaptation to each long-horizon task within the **LIBERO-10** benchmark over 50 epochs. Each experiment is conducted with 3 different random seeds. Except for the Experience Replay (ER) method, data from earlier tasks remains unavailable in later stages. Our diverse adaptation setup provides a thorough

---

[1] We use one adapter per task *suite*. LIBERO (Liu et al., 2023a) originally evaluated on a per-task basis.

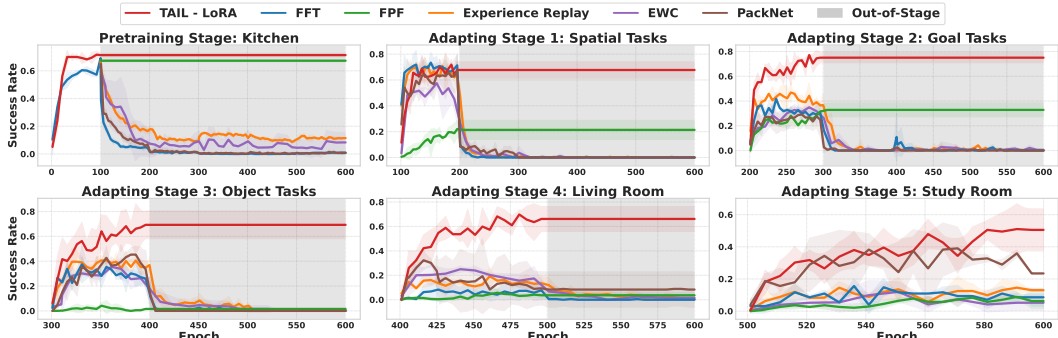

Figure 5: Success rates on the pretraining stage on 40 tasks in the LIBERO Kitchen scene and 5 adaptation stages, each with 8 tasks over 100 epochs, which are continuously evaluated in subsequent stages (shaded area).

and in-depth examination of knowledge transfer across a spectrum of domains, including spatial, procedural, visual, and compositional.

In the pretraining phase for TAIL, we add trainable adapters to the CLIP spatial and instruction encoders while freezing the encoder weights. All other model weights are fully learnable. During adaptation, the CLIP encoders and the GPT2 decoder are frozen, while adapters for them, the fusion module, and the policy head are tuned. Adapter weights are initialized from previous adapters with minor random noise. A fusion module and policy head copy are maintained during the adaptation for both TAIL and FPF. The detailed hyperparameters are presented in Appendix B.

## 5.3 RESULTS AND ANALYSIS

**Comparison of TAIL Integration Styles.** Fig. 4 showcases the continual adaptation success rates for different TAIL methods. The efficacy of LoRA suggests that a well-pretrained model has a surprisingly low intrinsic dimension for imitation learning tasks (Aghajanyan et al., 2020). This implies the existence of a low-rank reparameterization that is just as adept for fine-tuning as the full parameter space. Further, the prefix tuning method outperforms the bottleneck-based approach (Houlsby et al., 2019), indicating that the sequential integration style may not be the optimal choice for continual learning, potentially due to its inherent "filtering" mechanism. Surprisingly, RoboAdapter (Sharma et al., 2023) generally performs the worst, potentially due to only introducing weights after the feedforward layer as opposed to after (Houlsby et al., 2019) or within (Li & Liang, 2021; Hu et al., 2021) the attention layer. Due to LoRA's pronounced effectiveness, it is predominantly employed as our TAIL integration method in subsequent experiments.

**TAIL vs. Conventional Fine-tuning.** Across all evaluations, TAIL vastly outperforms all baselines in both forward and backward transfer, demonstrating that conventional fine-tuning methods are weak in data-scarce continual learning. In Fig. 5 we plot continual learning success rates over 6 task suites, where TAIL outperforms the best baselines by over **3x** in some comparisons and generally achieves the best success rates. We display additional results on LIBERO-10, long-horizon tasks, in Table 1. Here, TAIL again performs best, with perfect backward transfer and forward transfer capabilities significantly better than the baselines: FFT not only exhibits marked catastrophic forgetting of earlier tasks—evidenced by poor BWT—but also compromises the model's adaptability to new tasks. This decline in forward transfer is characterized by a steady descent in success rates as training progresses, displayed in Appendix Table 7. Such deterioration in flexibility has been recognized in other studies as well (Lyle et al., 2022; Kumar et al., 2023). PackNet is able to adapt well on some task suites as it learns new parameters within different parts of the model, but overall is still outperformed by TAIL.

Table 1: Adaptation results on 10 long horizon tasks, higher is better. The BWT for TAIL methods are all 0 (no forgetting). FPF results were omitted due to its near-zero performance. See per-task results in Appendix Table 7.

| | Conventional Fine-Tuning Methods | | | | | | TAIL-based Methods (**Ours**) | | | |
| | Full Fine-Tuning | | Experience Replay | | EWC | | LoRA | Prefix | Bottleneck | RoboAdapter |
| | FWT ↑ | BWT ↑ | FWT ↑ | BWT ↑ | FWT ↑ | BWT ↑ | FWT ↑ | FWT ↑ | FWT ↑ | FWT ↑ |
|---|---|---|---|---|---|---|---|---|---|---|
| Average | 0.48 ± 0.10 | -0.55 ± 0.21 | 0.45 ± 0.09 | -0.49 ± 0.23 | 0.30 ± 0.16 | -0.43 ± 0.20 | **0.70 ± 0.10** | 0.51 ± 0.15 | 0.46 ± 0.11 | 0.42 ± 0.13 |

**Adaptation Plasticity.** Exhaustive fine-tuning on specialized domains has been found to distort pretrained features (Kumar et al., 2022), undermining model adaptability. Our circle-back experiments

Table 3: Comparison of trainable parameters and memory usage for TAIL and FFT. We use (·%) and ↓ (·%) to denote the percentage of trainable parameter and the decrease of GPU memory w.r.t FFT.

| Method | Conventional | TAIL-based Methods (Ours) | | | |
|---|---|---|---|---|---|
| | Full Fine-Tuning | LoRA | RoboAdapter | Bottleneck Adapter | Prefix Tuning |
| CLIP (Spatial & Task Encoder) | 149.62M | 0.49M | 1.29M | 1.31M | 0.58M |
| GPT2 (Temporal Encoder) | 21.78M | 0.69M | 0.40M | 0.40M | 0.24M |
| Fusion module and policy head | 0.84M | 0.84M | 0.84M | 0.84M | 0.84M |
| Total Parameters | 172.24M | 2.02M (1.17%) | 2.53M (1.47%) | 2.55M (1.48%) | 1.66M (0.93%) |
| GPU Memory (Batch 14) | 20.1G | 15.5G (↓ 23%) | 14.0G (↓ 30%) | 14.9G (↓ 26%) | 15.8G (↓ 21%) |

in Table 2, where a full fine-tuned model is re-trained on prior task suites, demonstrate a steep performance drop upon re-visiting previously learned tasks. Additional experiments in Appendix C.3 further highlight this issue.

The training and validation losses, detailed in Appendix C.1 and Fig. 7, highlight FFT's propensity to overfit. This translates to a notable decline in success rates, reinforcing the challenges FFT faces in balancing retention of prior tasks with the assimilation of new ones.

While ER and the regularization-based method EWC exhibit some potential in mitigating catastrophic forgetting, they were detrimental to forward transfer performance. Their downsides are also reflected in storage and computing costs: ER requires more storage for previous data than TAIL LoRA adapter weights (e.g., Kitchen dataset at 28GB vs 7.8MB for TAIL's LoRA adapter). Furthermore, EWC presents significant challenges for larger models because of the increased GPU memory consumption from maintaining a copy of the entire weights of the old model in memory. We also found it to exhibit unstable training due to the regularization loss. More discussions are presented in Appendix B.1.

Table 2: The success rate of initial training and revisiting previous tasks with FFT. FFT suffers from catastrophic forgetting and performs worse on re-visits despite re-training on the same data.

| Type | LIBERO Task Suite | | |
|---|---|---|---|
| | Spatial | Goal | Object |
| Initial | 0.79 | 0.42 | 0.42 |
| Re-visit | 0.53 (↓ 0.26) | 0.20 (↓0.22) | 0.27 (↓0.15) |

**When does TAIL work best?** The efficacy of TAIL hinges significantly on the base model's features. We compare TAIL under different pretraining strategies and models in Appendix Sec. C.2 and C.3. In short, TAIL works best with our pretraining architecture and frozen CLIP visual/language encoders, and performance drops when we fine-tune the pretrained encoders, likely as FFT contaminates the rich CLIP features when fine-tuned in a niche domain with sparse data.

**Analysis Summary.** We argue in favor of a large pretrained base model augmented with numerous lightweight plugins tailored for different downstream tasks. This framework, TAIL, holds considerable promise for advancing embodied intelligence in real-world applications; the storage footprint of our entire model is about 660MB, and duplicating this model for each task in a stream of oncoming tasks is impractical. Meanwhile, the space occupied by one such model can accommodate as many as 84 task-specific adapters, which, as our experiments show, can outperform full fine-tuning regardless. Moreover, the features of the pretrained weights remain intact, ensuring their applicability across a broad array of domains. In summary, TAIL offers a promising avenue for the efficient adaptation of large decision-making models. Despite the fact that our method requires significantly less computation and memory (and storage), our experiments show that it consistently outperforms all prior approaches in the continual learning setting. We would also like to highlight that the TAIL framework is not restricted to imitation learning, but also other learning methods such as reinforcement learning.

## 6  CONCLUSION

In this study, we examined the challenges of efficiently adapting large pretrained models for decision-making and robotics applications. We proposed TAIL, an efficient adaptation framework for pretrained decision-making models. Through a comprehensive exploration of parameter-efficient fine-tuning (PEFT) techniques in TAIL, especially Low-Rank Adaptation (LoRA), we demonstrated their potential in enhancing adaptation efficiency, mitigating catastrophic forgetting, and ensuring robust performance across diverse tasks. Our empirical evaluations on the LIBERO benchmark further underscored the advantages of these techniques in continual learning scenarios. As the demand for adaptive, intelligent agents grows across various domains, the insights from this research offer a promising direction for the future of efficient model adaptation in decision-making contexts.

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

# Appendix: TAIL: Task-specific adapters for imitation learning with large pretrained models

## A  Model Architecture Details

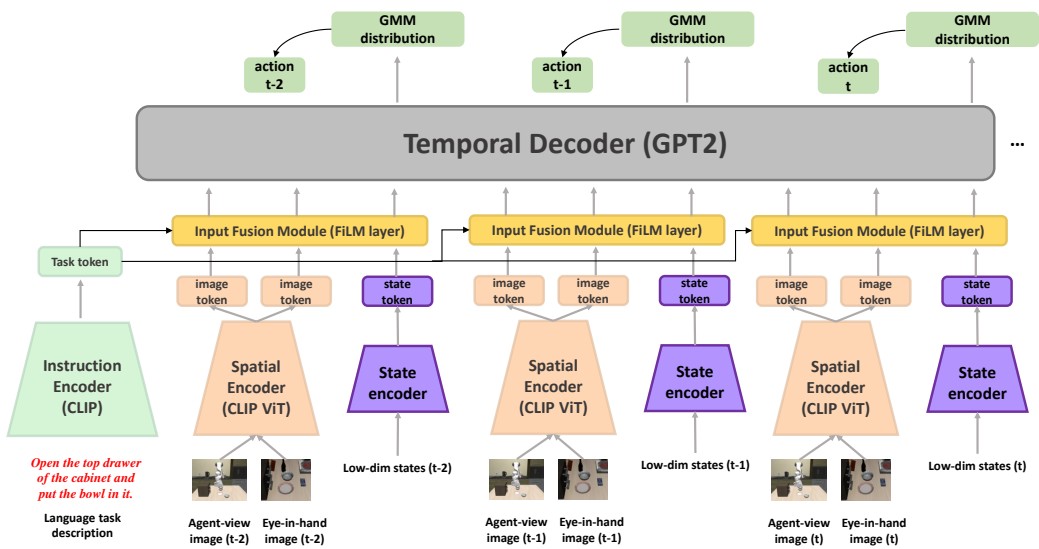

Figure 6: A detailed view of the multi-modal, transformer policy architecture we utilize for pretraining. We encode language task descriptions with a pretrained CLIP instruction encoder and image observations with a pretrained CLIP spatial encoder. We additionally encode robot state observations which, along with the observation embeddings, are embedded into a sequence of tokens used by the temporal decoder transformer to predict single-step action distributions. We include an input fusion module (FiLM (Perez et al., 2017)) to explicitly combine the task embedding with the observation and state embeddings for better instruction-following ability.

### A.1  Pretrained Input Encoders

We utilize pretrained CLIP image and textual encoders (Radford et al., 2021) to encode image observations and language goal descriptions, respectively. Note that we do not use a pre-trained encoder for the low-dimensional state; the state encoder is learned from scratch.

### A.2  Input Modality Fusion

We utilize Feature-wise Linear Modulation (FiLM) layers (Perez et al., 2017) (Fig. 1(a), input fusion module) to fuse language task specifications with image observations. FiLM is a technique in multi-modal deep learning which modulates the intermediate activations of a neural network based on external information. Rather than explicitly designing architectures for conditional computation, FiLM layers simply use the features from one network to modulate the features of another.

Let's consider a neural network $f$ with intermediate activations $x$ and an external network $g$ which outputs modulation parameters $\gamma$ and $\beta$. The modulated features $x'$ are given by:

$$\gamma, \beta = g(z) \tag{4}$$
$$x' = \gamma \odot x + \beta, \tag{5}$$

where $z$ is the input to the external network $g$; $\odot$ represents element-wise multiplication; $\gamma$ and $\beta$ are vectors having the same size as $x$, with each element modulating a corresponding feature in $x$.

Thus, FiLM layers allow for a dynamic and feature-wise conditional computation without needing explicit architectural changes. As such, task token (language) embeddings are given as input to a fully connected feedforward network, which outputs scale and translation parameters for the image and state embeddings. These parameters modulate the image and state embeddings before they are passed to the transformer backbone.

### A.3 Temporal Transformer Backbone

We utilize a standard GPT-2 (Radford et al., 2019a) transformer backbone for our policy. Its input is a sequence of image and low-dim state encodings (robot joint states in LIBERO) and it outputs an action distribution. Following the literature (Mandlekar et al., 2021; Liu et al., 2023a), we adopt a stochastic policy parametrization based on a Gaussian-Mixture-Model (GMM) (Bishop, 1994). Therefore, for every decision-making step, the transformer produces a latent vector of Gaussian means and variances, one for each of the GMM modes. We optimize the parameters of the model with the negative log-likelihood loss on the ground truth actions based on the parameters of the GMM distribution. At evaluation time, we deterministically select the next action parameterized by the mean of the Gaussian model with the highest density.

The environment configuration and the temporal decoder (GPT-2) hyperparameters are presented in Table 4.

Table 4: Environment configuration and GPT-2 model hyperparameters

| Environment Configuration | | GPT2 Temporal Encoder Configuration | | | |
|---|---|---|---|---|---|
| Action Dim. | 7 | Max Seq Length | 8 | Activation | Gelu New |
| Raw State Dim. | 9 | Number of Heads | 8 | Number of Layers | 6 |
| Max Episode Length | 500 | GMM Min Std | 0.0001 | GMM Modes | 5 |
| Image Resolution | 128 x 128 | FiLM Layers | 2 | Dropout | 0.15 |
| Image Views | Agent Front, Eye-in-Hand | | | | |

## B  Implementation and Training Details

### B.1  Baseline Details

**Experience Replay (ER).** ER (Chaudhry et al., 2019; Rolnick et al., 2019) is a rehearsal-based approach that retains a buffer of samples from previous tasks to facilitate the learning of new tasks. After completing the learning process for a task, a subset of the data is saved into this buffer. During the training of subsequent tasks, ER draws samples from this buffer and mixes them with current task data. This process ensures that the training data closely resembles the distribution of data across all tasks. In our setup, we store all the previous trajectories in a replay buffer. For each training iteration on a new task, we uniformly sample $50\%$ trajectories from this buffer and $50\%$ from the new task's training data, respectively.

**Elastic Weight Consolidation (EWC).** EWC (Kirkpatrick et al., 2017) is a regularization method that adds a term to the standard single-task learning objective to constrain the updates of the neural network. This constraint uses the Fisher information matrix to gauge the significance of each network parameter. The loss function for task $k$ is represented as:

$$L_{\text{EWC}_k}(\theta) = L_{\text{BC}_K}(\theta) + \sum_i \frac{\lambda}{2} F_i (\theta_i - \theta_{k-1,i}^*)^2$$

Here, $\lambda$ is a hyperparameter penalty, and $F_i$ is the diagonal of the Fisher information matrix given by:

$$F_k = \mathbb{E}_{s \sim D_k, a \sim p_\theta(\cdot|s)} \left( \nabla_{\theta_k} \log p_{\theta_k}(a|s) \right)^2$$

For our experiments, we adopt the online version of EWC. It updates the Fisher information matrix using an exponential moving average throughout the lifelong learning process. The actual Fisher Information Matrix estimate used is:

$$\tilde{F}_k = \gamma F_{k-1} + (1 - \gamma) F_k$$

with $F_k = \mathbb{E}_{(s,a) \sim D_k} \left( \nabla_{\theta_k} \log p_{\theta_k}(a|s) \right)^2$ and $k$ representing the task number. Following the benchmark implementation (Liu et al., 2023a), the hyperparameters are set as $\gamma = 0.9$ and $\lambda = 5 \times 10^4$.

**Discussions.** Both Experience Replay (ER) and Elastic Weight Consolidation (EWC) demonstrate potential in mitigating catastrophic forgetting. However, they each come with notable limitations, particularly with respect to forward transfer performance, storage, and computational efficiency.

*Storage Overhead:* ER demands significant storage space to maintain samples from prior tasks. This becomes particularly evident when comparing the storage needs of ER for larger datasets, such as the Kitchen dataset which requires 28GB, with the lightweight LoRA adapter occupies only 7.8MB. The vast difference in storage demands underscores the inefficiency of the ER approach.

*Computational Challenges:* EWC, by design, necessitates the maintenance of a copy of the weights of the previous model in GPU memory. This leads to escalated GPU memory consumption, making EWC tends to reduce the training batch size, subsequently slowing down the training process.

*Training Instability:* The regularization approach of EWC can introduce instability during training, owing to the regularization loss. This is also reflected by the poor forward transfer capability, as shown in Table 1.

*Scalability Concerns:* While EWC might be manageable for smaller networks, it is ill-suited for the fine-tuning of larger decision models due to its computational and storage challenges.

Given these outlined limitations, we advocate TAIL for alternative approaches that are both storage-efficient and computationally scalable, especially for large pretrained model adaptation.

### B.2 TAIL ADAPTER CONFIGURATIONS

To establish our TAIL adapter configurations, we primarily draw from the AdapterHub implementation, setup and hyperparameters (Pfeiffer et al., 2020b).

We utilize the default hyperparameters for LoRA, with the rank $r = 8$ and scaling factor $\alpha = 8$. These low-rank matrices are applied in parallel to the Transformer's query and value matrices (Hu et al., 2021). We also adopt the default for prefix token length of 30 for the prefix tuning (Li & Liang, 2021) method across all tasks. To improve the training stability, Low-rank matrices ($r = 16$) are employed during training to represent the prefix tokens. The Bottleneck Adapter (Houlsby et al., 2019) employs the bottleneck size of 32, and is applied to both the output layer of the attention and the intermediate feedforward layers. The RoboAdapter method (Sharma et al., 2023), as the closest work to us, also applies the sequential adapters to the decision-making domain. It differs from the Bottleneck Adapter in that they adopt a special insertion of weights to specific layers of the Transformer, namely, layers $0, 1, 5, 6, 10, 11$. They selectively skip certain layers, aiming to increase the bottleneck size on the remaining layers. Therefore, the bottleneck size is doubled to 64 for this approach, such that all methods share similar amount of parameters.

In order to maintain balanced adapter parameters number between the two CLIP-based (spatial and instruction) encoders, and the temporal transformer GPT2 decoder, the rank size for the GPT2 decoder is doubled across all methodologies. This adjustment compensates for the GPT2 decoder's fewer layers relative to the encoders.

For the continual learning setup, we use the previous stage's adapter weight (if any) plus a small random Gaussian noise with standard deviation 0.001 as an initialization of the current stage. The goal for adding a minor random noise aims to improve the adapter weight capacity (Kumar et al., 2022; Agarwal et al., 2022; Lyle et al., 2022), preventing the weights from being trapped into local optimum. There is a potential to better utilize the trained adapter weights on preceding tasks. We outline several promising exploration directions in Appendix Section B.4.

### B.3 TRAINING HYPERPARAMETERS AND EXPERIMENT CONFIGURATIONS

Following similar setup as in the LIBERO benchmark (Liu et al., 2023a), we perform data augmentation for the image observation data for all methods. We adopt the color, affine, and random erase augmentations to improve the robustness. The hyperparameters are presented in Table 5.

For our training process, we employed the AdamW optimizer combined with a linear learning rate scheduler. The majority of our task suites—Kitchen, Spatial, Goal, Object, Living Room, and Study Room—underwent training for 100 epochs. Notably, each suite encompasses multiple tasks, with Kitchen having 40 and the others containing 8 each. In contrast, the 10 long-horizon adaptation tasks,

Table 5: Image data augmentation and training hyperparameters

| Image Augmentation | | | | Training and Optimizer Configuration | | | |
|---|---|---|---|---|---|---|---|
| Brightness | 0.3 | Contrast | 0.3 | Training Epochs | 100/50 | Batch Size (per device) | 10/14/18 |
| Saturation | 0.3 | Hue | 0.3 | Training Epochs per Eval | 5 | Eval Episodes/Task | 8 |
| Color Aug Prob. | 0.9 | Affine Degrees | 15 | Warm-up Steps | 500 | Weight Decay | 0.1 |
| Affine Translate | 0.1 | Affine Prob. | 0.6 | Learning Rate (LR) | 1e-4 | LR Scheduler | Linear |
| Random Erase Prob. | 0.1 | | | Training Demo Num | 40 | Validation Demo Num | 40 |

termed LIBERO-10, were trained for 50 epochs, with each task trained sequentially. We performed evaluations after every 5 training epochs over 8 episodes (unseen in training) for each task.

**Computing machine.** Our experimental platform was powered by an AMD EPYC 7R32 CPU running Ubuntu 20.04.06. All trainings utilized 8 NVIDIA A10G GPUs, each with a memory of 22731 MiB, equipped with driver version 470.199.02 and CUDA version 11.4. We employ Distributed Data Parallel (DDP) for parallel training across 8 GPUs, and utilize the 16-bit floating point precision (FP16) training mode to accelerate the training process. To ensure reproducibility, we adopted 3 distinct random seeds: 0, 21, and 42.

**Training time.** For a holistic perspective on training duration: FFT and ER methods demanded between $120 \sim 140$ hours per experiment ($1.5 \sim 1.75$ hours per task) for the 6 task suites shown in Fig. 5, including the evaluation time. In stark contrast, TAIL-based techniques slashed this to $60 \sim 66$ hours ($0.75 \sim 0.825$ hours per task). Hence, TAIL would also be much cheaper to train, considering its significantly shorter training time under identical computing machines.

Batch sizes varied by training method. EWC employed a batch size of 10, given its added memory demands to store a distinct full parameter set. FFT and ER utilized batch sizes of 14. Owing to TAIL's more efficient memory utilization—detailed in Table 3—a larger batch size of 18 was feasible, which can maximize GPU resource usage on our machine, reducing training duration and cost.

### B.4 MORE DISCUSSION AND FUTURE DIRECTIONS

The TAIL framework paves the way for a myriad of research opportunities:

1. **Better Weight Allocation Method Across Layers:** An interesting question within this framework is discerning which layers, early or later, derive the most benefit from weight modifications. This can offer insights into the adaptability of neural architectures (Lee et al., 2023).

2. **Enhanced Reusability of Trained Adapters:** Exploring methods to efficiently reuse adapters from prior tasks, especially in scenarios with limited data, is a promising direction. AdapterFusion techniques (Pfeiffer et al., 2020a) can be potentially useful, enabling the composition of knowledge from multiple pre-existing adapters.

3. **Building on Knowledge with Parallel Integration:** The parallel integration method, particularly with LoRA weights, offers the capability to merge trained weights back into the main model. This iterative buildup of knowledge makes the approach valuable for continual learning, allowing new adapters to capitalize on the expertise of their predecessors.

4. **Combining with Established Continual Learning Strategies:** The potential to merge the TAIL framework with existing continual learning methods, like Experience Replay and EWC, can be a beneficial avenue. Such integrations can accommodate the strengths of each method, crafting models that are both efficient in memory and robust against forgetting.

5. **Extension beyond the Imitation Learning Domain:** Taking the TAILframework into other decision-making domains, such as reinforcement learning (RL), is also promising. TAIL has the potential to address the model capacity loss issue in RL (Agarwal et al., 2022; Lyle et al., 2022). Leveraging the TAIL framework can also aid in multitask learning, meta-learning, and efficiently adapting offline-trained RL models to new tasks without the necessity of vast amounts of data or extensive fine-tuning, thereby potentially accelerating convergence to optimal policies.

The avenues above elucidate the adaptability and potential of the TAIL framework, setting the stage for future research in this domain.

## C    MORE EXPERIMENT RESULTS

### C.1    OVERFITTING

For each task, we used 40 demonstrations for training and 10 for validation. We are interested in the following question: *In scenarios where data is limited, how resilient is TAIL against overfitting compared to traditional fine-tuning methods?* To answer this, we present the training and validation loss cross the Kitchen, Spatial, Goal, Object, Living Room and Study Room task suites, each with 100 epochs, in Fig. 7.

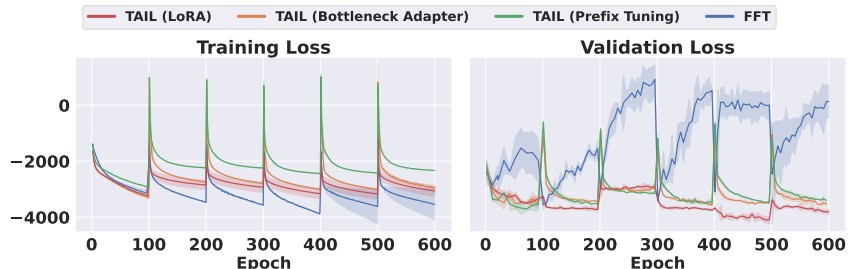

Figure 7: Adaptation loss trends: Training versus validation. The graph shows that the TAIL model consistently has more stable validation losses, which means that it is more robust to contexts with limited data. On the other hand, the full fine-tuning model (FFT) has larger validation losses, which means that it is more likely to overfit to the training data.

A noteworthy observation from Fig. 7 is the behavior of FFT. Despite achieving the lowest training loss across all stages, its validation loss spikes significantly after just a few epochs. This pattern suggests severe overfitting when FFT is applied to the entire parameter space using limited data. Intriguingly, this overfitting intensifies in the later adaptation phases, potentially signifying a distortion of pretrained features as alluded to by Kumar et al. (2022). Such distortion could be a contributor to the suboptimal success rate observed in Fig. 5, and the loss of learning capacity when revisiting a previous task, as presented in Table 2.

In constrast, TAIL-based methods shows strong resilience against overfitting. Drawing from the Occam's razor principle, TAIL leverages fewer trainable parameters, inherently reducing its potential to overfit with scarce data. Additional, different integration styles provide the flexibility to effectively utilize the features from pretrained models while preserving them across all the adaptation stages.

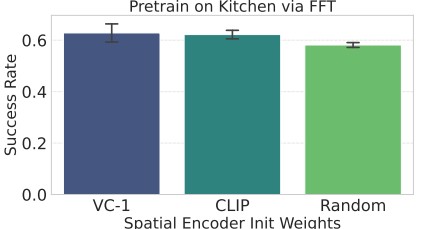

Figure 8: Training on the Kitchen task with different pretrained CLIP-ViT encoder weight. Random means using random initialization weight.

This observation underscores the disparities between our decision-making problem, characterized by its limited data, and the traditional language or vision domains, which have data in abundance. Prior studies utilizing parameter-efficient fine-tuning techniques for language or vision tasks often reported superior performance with full fine-tuning due to its low training loss (He et al., 2022; Mao et al., 2022; Chen et al., 2022; Sharma et al., 2023). However, as our results demonstrate, a lower training loss does not invariably translate to superior performance, especially in the context of a data-scarce sequential decision-making tasks.

### C.2    ANALYSIS OF PRETRAINED WEIGHTS' INFLUENCE

We aim to answer the following question: *how does the underlying pretrained base model influence the performance of TAIL, and are certain pretrained weights more conducive to this kind of adaptation?* We initiated our investigation by analyzing the success rates of 40 Kitchen tasks using different pretrained weights for the spatial encoder. Apart from the CLIP-ViT pretrained encodings as we adopted in our main results, two other initialization of weights were considered: one sourced from the Visual Cortex 1 (VC-1) (Majumdar et al., 2023b), recognized for being a leading pretrained

model for embodied agent tasks, and another using randomly initialized weights. The language instruction encoder consistently utilized the CLIP text model. From the results in Fig. 8, the VC-1 pretrained weights delivered performance on par with the CLIP-ViT encodings. Both considerably outperformed the randomly initialized weights, suggesting that large-scale pretraining can indeed enhance downstream fine-tuning. We then study how does the pretrained base model influence the performance of TAIL.

## C.3 FURTHER EVALUATIONS ON TAIL WITH DIFFERENT BASE MODELS

To understand the influence of the base model's features on the performance of TAIL, we conducted additional evaluations. In Table 6, the methods column showcases different configurations:

- **LoRA (CLIP):** The main setup we adopted in the experiment section 5, which keeps the pretrained CLIP encodings frozen across all the adaptation stages.
- **LoRA (CLIP with FFT):** Starting with the CLIP model, we applied FFT pretraining on the Kitchen task before using LoRA for subsequent adaptations. This helps us test out whether adaptation plasticity suffers after full fine-tuning as the only difference between this and the above method is the addition of full fine-tuning before using LoRA.
- **LoRA (VC-1 with FFT):** The VC-1 model, after FFT pretraining on the Kitchen task, was adapted using LoRA.
- **LoRA (Random with FFT):** A model with randomly initialized weights underwent FFT pretraining on the Kitchen task, followed by adaptation with LoRA.

All the pretrained encodings implemented in the same model architecture as described in Appendix Section A.

Observations from Table 6 highlight several findings:

- **Dominance of Original CLIP:** The pure CLIP base model, when combined with LoRA, yielded the highest success rates across all task suites, suggesting the inherent quality and robustness of the original CLIP features for these tasks.
- **FFT's Mixed Impact:** While FFT pretraining aids in task-specific fine-tuning, when combined with CLIP, it leads to a degradation in performance. This could be attributed to FFT potentially diluting the comprehensive and rich features within CLIP while also reducing adaptation plasticity (Kumar et al., 2022), especially when exposed to a more constrained domain with limited data.
- **VC-1's Comparable Performance:** The VC-1 model, though renowned in the domain of embodied agent tasks, delivered results that were only marginally better than the randomly initialized weights when both were subjected to FFT pretraining and then adapted with LoRA. This emphasizes the unique advantages of the original CLIP features.

Interestingly, it is observed that CLIP is pretrained on the most comprehensive dataset, followed by VC-1. In contrast, the model with random weights only underwent pretraining on the 40 Kitchen tasks. The success rates mirror this order, underscoring the idea that the efficacy of TAIL is closely tied to a base model pretrained with rich features on extensive datasets. So in summary, the choice of base model significantly affects the performance of TAIL, with CLIP's original features showing remarkable compatibility and resilience across various task suites

Table 6: Evaluation results of FWT for LoRA with different pretrained model weights. The higher, the better. We highlight the best method with highest FWT as **bold**.

| Method | Spatial | Goal | Object | Living Room | Study Room | Average |
|---|---|---|---|---|---|---|
| LoRA (CLIP) | **0.76** ±0.02 | **0.79** ±0.02 | **0.73** ±0.14 | **0.73** ±0.07 | **0.55** ±0.11 | **0.71** ±0.07 |
| LoRA (CLIP with FFT) | 0.62 ±0.04 | 0.67 ±0.13 | 0.38 ±0.08 | 0.32 ±0.08 | 0.32 ±0.01 | 0.46 ±0.07 |
| LoRA (Random with FFT) | 0.38 ±0.19 | 0.60 ±0.06 | 0.37 ±0.03 | 0.23 ±0.01 | 0.47 ±nan | 0.41 ±0.07 |
| LoRA (VC-1 with FFT) | 0.56 ±0.07 | 0.66 ±0.08 | 0.25 ±0.00 | 0.20 ±0.06 | 0.48 ±0.07 | 0.43 ±0.05 |

## C.4 RANK SIZE ABLATION STUDY

In order to understand the impact of rank-size on adaptation performance, we conducted experiments using varying rank sizes for the LoRA and Bottleneck Adapter methods. The results, illustrated in Fig. 9, present the average success rates across the Spatial, Goal, and Object task suites. It is evident that increasing the rank size generally enhances performance up to a certain point. Beyond this optimal threshold, further increasing the rank size does not necessarily lead to higher success rates, potentially because of overfitting. Notably, in our continual learning context, the parallel insertion approach of LoRA consistently surpasses the sequential style of the Bottleneck Adapter method.

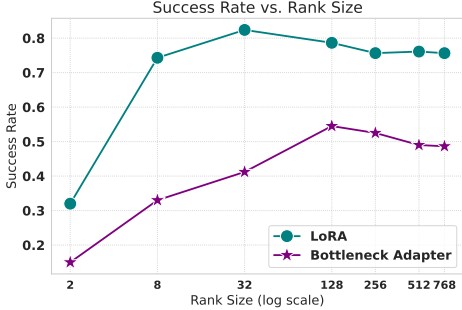

Figure 9: Ablation study of the rank-size of LoRA and Bottleneck adapters. Increasing the rank size generally enhances performance up to a certain point. Beyond this optimal threshold, further increasing the rank size does not necessarily lead to higher success rates. The parallel insertion approach of LoRA consistently surpasses the sequential style of the Bottleneck Adapter method

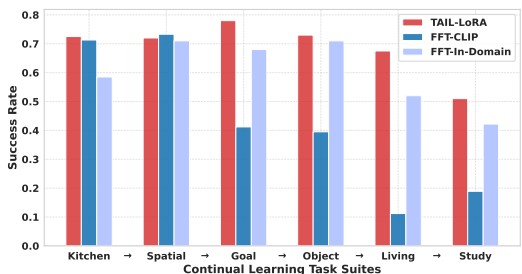

Figure 10: Comparison for TAIL-LoRA, sequential FFT with pre-trained CLIP weights, and FFT-In-Domain. FFT-In-Domain is trained from scratch with task-suite demonstration data only, which saves a copy of the entire model for each task. FFT with CLIP excels in initial Kitchen and Spatial suites, highlighting the value of pretrained models; however, its performance declines in subsequent tasks, suggesting reduced adaptability. In contrast, TAIL-LoRA demonstrates consistent superior performance across all suites.

Additionally, we would like to note that our TAIL framework exhibits data adaptivity, suggesting that the rank size could be adjusted based on the quantity of adaptation data. In scenarios with smaller datasets, a smaller rank size could be more effective, and vice versa.

## C.5 COMPARISON BETWEEN TRAINING FROM SCRATCH AND USING PRETRAINED MODELS

Fig. 10 compares the success rates across task suites for TAIL-LoRA, sequential FFT with pre-trained CLIP weights, and FFT-In-Domain. Unlike FFT-CLIP, FFT-In-Domain is trained from scratch with task-suite demonstration data only, i.e., we need to maintain a copy of the entire model for each task suite. There are three observations:

**1. Pretrained Weights Advantage:** In the initial Kitchen and Spatial task suites, FFT with CLIP pretrained weights demonstrates a higher success rate compared to FFT trained from scratch. This indicates the effectiveness of leveraging pretrained models, particularly in the context of the Kitchen suite where the benefit is more pronounced.

**2. Decline in Model Adaptability:** Despite the initial advantage, sequential FFT with CLIP shows a marked decline in performance in the remaining four task suites - Goal, Object, Living, and Study. This trend may be indicative of a loss in model plasticity, where the pre-trained model performs well in the early stages but struggles to adapt to new tasks after the pre-trained weights are contaminated.

**3. TAIL-LoRA's Consistent Performance:** Throughout all the task suites, TAIL-LoRA with pretrained CLIP consistently outperforms the other methods. This suggests that the LoRA approach, combined with the advantages of pretrained CLIP weights, provides a robust and adaptable framework capable of handling a variety of tasks with greater efficiency.

## C.6 ABLATION STUDY FOR DIFFERENT INTEGRATION STYLE COMBINATIONS

It's noteworthy that our method allows for the simultaneous use of multiple integration techniques (Mao et al., 2022). This flexibility lets us explore the performance impact of combining LoRA (parallel integration), bottleneck adapter (sequential integration), and prefix token (concatenation). To this end, we conduct an ablation study for each of the combinations over the Spatial, Goal, and Object task suites. The experiment result is shown in Fig. 11, where the y-axis is the averaged success rate.

A key finding is the critical role of LoRA (parallel integration) in enhancing adaptation performance. Combinations involving LoRA consistently outperform those without it. For instance, the standalone use of LoRA yields a comparable success rate w.r.t the combination with others. This pattern underscores LoRA's effectiveness, either used alone or in conjunction with other methods. In contrast, the combination of Prefix and Adapter without LoRA results in a notably lower success rate (0.6641), highlighting LoRA's indispensability.

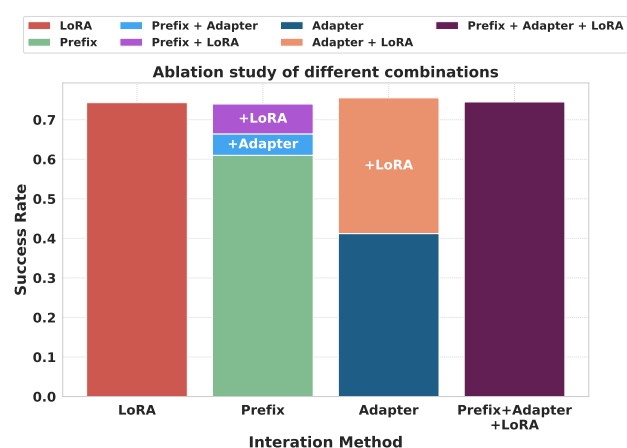

Figure 11: Ablation study for integration style combinations. LoRA (parallel integration) plays a crucial role in enhancing adaptation performance, consistently outperforming methods without it. Whether used alone or in combination with other methods like Prefix and Adapter, LoRA shows superior effectiveness.

The integration of all three methods—Prefix, Adapter, and LoRA—achieves a success rate that is comparable to LoRA's standalone performance. This outcome suggests that while the combination of different integration methods does not detract from performance, LoRA remains the primary driver of successful adaptation. These findings emphasize the importance of LoRA in adapter weight integration strategies and provide valuable guidance for future approaches in this domain.

## C.7 DETAILED PER-TASK RESULTS IN THE LIBERO-LONG TASK SUITE

Table 7: Adaptation results on 10 long horizon tasks. The ↑ symbol means the higher, the better. The BWT ↑ for TAIL methods are all 0 (no catastrophic forgetting). We highlight the best method (highest FWT ↑) in **bold**. FPF results were omitted due to its near-zero performance.

| Task | Conventional Fine-Tuning Methods | | | | | | TAIL-based Methods (**Ours**) | | | |
| | Full Fine-Tuning | | Experience Replay | | EWC | | LoRA | Prefix | Bottleneck | RoboAdapter |
| | FWT ↑ | BWT ↑ | FWT ↑ | BWT ↑ | FWT ↑ | BWT ↑ | FWT ↑ | FWT ↑ | FWT ↑ | FWT ↑ |
|---|---|---|---|---|---|---|---|---|---|---|
| Task 1 | $0.42 \pm 0.07$ | - | $0.25 \pm 0.12$ | - | $0.38 \pm 0.12$ | - | $\mathbf{0.62 \pm 0.00}$ | $0.38 \pm 0.12$ | $0.21 \pm 0.14$ | $0.12 \pm 0.00$ |
| Task 2 | $0.58 \pm 0.07$ | $-0.42 \pm 0.06$ | $0.58 \pm 0.07$ | $-0.25 \pm 0.10$ | $0.54 \pm 0.07$ | $-0.38 \pm 0.10$ | $\mathbf{0.75 \pm 0.00}$ | $0.58 \pm 0.19$ | $\mathbf{0.75 \pm 0.12}$ | $0.50 \pm 0.12$ |
| Task 3 | $0.71 \pm 0.07$ | $-0.50 \pm 0.10$ | $0.67 \pm 0.07$ | $-0.42 \pm 0.19$ | $0.38 \pm 0.12$ | $-0.46 \pm 0.12$ | $\mathbf{0.96 \pm 0.07}$ | $0.88 \pm 0.22$ | $0.71 \pm 0.19$ | $0.50 \pm 0.25$ |
| Task 4 | $\mathbf{0.96 \pm 0.07}$ | $-0.57 \pm 0.13$ | $0.92 \pm 0.07$ | $-0.50 \pm 0.20$ | $0.75 \pm 0.25$ | $-0.43 \pm 0.12$ | $0.88 \pm 0.00$ | $0.71 \pm 0.07$ | $0.71 \pm 0.19$ | $0.58 \pm 0.14$ |
| Task 5 | $0.21 \pm 0.07$ | $-0.67 \pm 0.21$ | $0.33 \pm 0.14$ | $-0.60 \pm 0.25$ | $0.17 \pm 0.19$ | $-0.50 \pm 0.18$ | $\mathbf{0.62 \pm 0.12}$ | $0.17 \pm 0.07$ | $0.25 \pm 0.00$ | $0.29 \pm 0.07$ |
| Task 6 | $\mathbf{0.83 \pm 0.19}$ | $-0.57 \pm 0.26$ | $0.71 \pm 0.19$ | $-0.55 \pm 0.25$ | $0.50 \pm 0.43$ | $-0.42 \pm 0.19$ | $0.75 \pm 0.12$ | $0.79 \pm 0.14$ | $0.75 \pm 0.00$ | $0.75 \pm 0.25$ |
| Task 7 | $0.17 \pm 0.07$ | $-0.62 \pm 0.27$ | $0.12 \pm 0.00$ | $-0.58 \pm 0.25$ | $0.04 \pm 0.07$ | $-0.44 \pm 0.24$ | $\mathbf{0.54 \pm 0.26}$ | $0.38 \pm 0.12$ | $0.31 \pm 0.09$ | $0.33 \pm 0.07$ |
| Task 8 | $0.42 \pm 0.07$ | $-0.55 \pm 0.29$ | $0.29 \pm 0.07$ | $-0.51 \pm 0.28$ | $0.12 \pm 0.18$ | $-0.46 \pm 0.28$ | $\mathbf{0.75 \pm 0.25}$ | $0.67 \pm 0.19$ | $0.25 \pm 0.18$ | $0.50 \pm 0.22$ |
| Task 9 | $0.17 \pm 0.07$ | $-0.54 \pm 0.28$ | $0.12 \pm 0.05$ | $-0.50 \pm 0.28$ | $0.00 \pm 0.00$ | $-0.41 \pm 0.29$ | $\mathbf{0.38 \pm 0.12}$ | $0.08 \pm 0.07$ | $0.19 \pm 0.09$ | $0.21 \pm 0.07$ |
| Task 10 | $0.33 \pm 0.19$ | $-0.50 \pm 0.29$ | $0.50 \pm 0.02$ | $-0.46 \pm 0.29$ | $0.12 \pm 0.18$ | $-0.38 \pm 0.31$ | $\mathbf{0.79 \pm 0.07}$ | $0.50 \pm 0.33$ | $0.44 \pm 0.09$ | $0.42 \pm 0.07$ |
| Average | $0.48 \pm 0.10$ | $-0.55 \pm 0.21$ | $0.45 \pm 0.09$ | $-0.49 \pm 0.23$ | $0.30 \pm 0.16$ | $-0.43 \pm 0.20$ | $\mathbf{0.70 \pm 0.10}$ | $0.51 \pm 0.15$ | $0.46 \pm 0.11$ | $0.42 \pm 0.13$ |

## D EVALUATION TASK DETAILS

We list all the language instructions describing the tasks we adopted in our experiments below. Note that while certain tasks may share similar descriptions, they are not the same due to variations in the environment configurations (e.g., different spatial layouts, objects, or goal positions).

| Task Suite | Instructions |
|---|---|
| Kitchen | close the top drawer of the cabinet |
| | close the top drawer of the cabinet and put the black bowl on top of it |
| | put the black bowl in the top drawer of the cabinet |
| | put the butter at the back in the top drawer of the cabinet and close it |
| | put the butter at the front in the top drawer of the cabinet and close it |
| | put the chocolate pudding in the top drawer of the cabinet and close it |
| | open the bottom drawer of the cabinet |
| | open the top drawer of the cabinet |
| | open the top drawer of the cabinet and put the bowl in it |
| | put the black bowl on the plate |
| | put the black bowl on top of the cabinet |
| | open the top drawer of the cabinet |
| | put the black bowl at the back on the plate |
| | put the black bowl at the front on the plate |
| | put the middle black bowl on the plate |
| | put the middle black bowl on top of the cabinet |
| | stack the black bowl at the front on the black bowl in the middle |
| | stack the middle black bowl on the back black bowl |
| | put the frying pan on the stove |
| | put the moka pot on the stove |
| | turn on the stove |
| | turn on the stove and put the frying pan on it |
| | close the bottom drawer of the cabinet |
| | close the bottom drawer of the cabinet and open the top drawer |
| | put the black bowl in the bottom drawer of the cabinet |
| | put the black bowl on top of the cabinet |
| | put the wine bottle in the bottom drawer of the cabinet |
| | put the wine bottle on the wine rack |
| | close the top drawer of the cabinet |
| | put the black bowl in the top drawer of the cabinet |
| | put the black bowl on the plate |
| | put the black bowl on top of the cabinet |
| | put the ketchup in the top drawer of the cabinet |
| | close the microwave |
| | put the yellow and white mug to the front of the white mug |
| | open the microwave |
| | put the white bowl on the plate |
| | put the white bowl to the right of the plate |
| | put the right moka pot on the stove |
| | turn off the stove |

Table 8: 40 Kitchen scene pretraining tasks

| Task Suite | Instructions |
|---|---|
| Long-horizon (LIBERO 10) | put both the alphabet soup and the tomato sauce in the basket
put both the cream cheese box and the butter in the basket
turn on the stove and put the moka pot on it
put the black bowl in the bottom drawer of the cabinet and close it
put the white mug on the left plate and put the yellow and white mug on the right plate
pick up the book and place it in the back compartment of the caddy
put the white mug on the plate and put the chocolate pudding to the right of the plate
put both the alphabet soup and the cream cheese box in the basket
put both moka pots on the stove
put the yellow and white mug in the microwave and close it |
| Spatial | pick up the black bowl between the plate and the ramekin and place it on the plate
pick up the black bowl next to the ramekin and place it on the plate
pick up the black bowl from table center and place it on the plate
pick up the black bowl on the cookie box and place it on the plate
pick up the black bowl in the top drawer of the wooden cabinet and place it on the plate
pick up the black bowl on the ramekin and place it on the plate
pick up the black bowl next to the cookie box and place it on the plate
pick up the black bowl on the stove and place it on the plate |
| Goal | open the middle drawer of the cabinet
put the bowl on the stove
put the wine bottle on top of the cabinet
open the top drawer and put the bowl inside
put the bowl on top of the cabinet
push the plate to the front of the stove
put the cream cheese in the bowl
turn on the stove |
| Object | pick up the alphabet soup and place it in the basket
pick up the cream cheese and place it in the basket
pick up the salad dressing and place it in the basket
pick up the bbq sauce and place it in the basket
pick up the ketchup and place it in the basket
pick up the tomato sauce and place it in the basket
pick up the butter and place it in the basket
pick up the milk and place it in the basket |
| Living Room | pick up the alphabet soup and put it in the basket
pick up the butter and put it in the basket
pick up the milk and put it in the basket
pick up the orange juice and put it in the basket
pick up the tomato sauce and put it in the basket
pick up the alphabet soup and put it in the tray
pick up the butter and put it in the tray
pick up the cream cheese and put it in the tray |
| Study Room | pick up the book and place it in the right compartment of the caddy
pick up the book and place it in the front compartment of the caddy
pick up the book and place it in the left compartment of the caddy
pick up the book and place it in the right compartment of the caddy
pick up the red mug and place it to the right of the caddy
pick up the white mug and place it to the right of the caddy
pick up the book in the middle and place it on the cabinet shelf
pick up the book on the left and place it on top of the shelf |

Table 9: Adaptation task suites

