# OpenReview forum: "TAIL: Task-specific Adapters for Imitation Learning with Large Pretrained Models"
_ICLR.cc/2024/Conference — ICLR 2024 poster_

### Official Review · Reviewer_Lua3 · 2023-10-23

**Soundness:** 3 good
**Presentation:** 4 excellent
**Contribution:** 2 fair
**Rating:** 8
**Confidence:** 4

**Summary:**

The work presents Task-specific Adapters for Imitation Learning (TAIL), a framework for fine-tuning control policies in continual learning setting by borrowing ideas from large language model space. The agent is trained on multiple tasks, one task at the time, and the objective is to learn the new tasks while retaining the performance in old tasks. TAIL algorithms update only a small set of parameters, inspired by parameter efficient training from language model space. The results show that TAIL combined with LoRA outperforms all tested alternatives, especially full finetuning (tuning the whole network) or other methods that fine-tune the whole network.

**Strengths:**

Overall nicely wrapped work. While not the most novel solution (see weaknesses), I feel the amount of results and experiments conducted here merit acceptance. I believe these results will be useful for many readers, and the instructions on how to fine-tune these models will support future work.

### Originality

Authors bring insights from language space to control space in the form of efficient finetuning algorithms, and test those algorithms in a new setting (continual learning).

### Quality

Experiments are throughout with enough ablations and baselines. The methods are evaluated from different angles (performance increase in new task, retainibility in previous tasks, number of parameters / computational requirements.)

Accurate description of the experiment setup provided in the appendix (e.g., hardware, driver versions).

### Clarity

The manuscript is written well and is clear to read.

### Significance

The insights are also helpful for simple fine-tuning of models for different tasks: results show that LoRA is also very fast to adapt in general (better than full finetuning).

The manuscript also provides a list of ablations on what settings are recommended for finetuning, which are useful for people to follow.

Given the popularity of transformer models and their adaptation in different domains, I see this work being useful for many readers. Focus on experiments that can be ran on lesser hardware (e.g., single GPU, less space) also allows more people to experiment with these features.

**Weaknesses:**

- Using a separate set of weights ("adapters") per task feels bit of cheating when comparing the model against the baselines. This alone ensures that you keep high performance in the previous tasks. While the argument for this is valid (with TAIL, you only need a handful of parameters vs. full network), I'd prefer a more apples-to-apples comparison.
- Limited to only one environment. While I understand data for right setup is hard to come by, it is hard to tell if the results generalize outside this tested environment and data. Showing results in an another type of environment would solidify these insights.
- (Minor) Proposed framework, despite having its own name, is not especially novel or specific: it is an umbrella term for existing solutions applied on the continual learning setting. This is evident from figures, where the same method (TAIL) appears multiple times, and fundamentally, all these variants are significantly different from each other. I'd perhaps call different instiations of this setup "TAIL-LoRA", instead of "TAIL (LoRA)", to signify it is LoRA applied in TAIL's fashion.

**Questions:**

1) Page 7, "Training, Adaptation and Evaluation" paragraph: Manuscript says "This limited data setup...". How do you consider this a "limited data" setup? Can you give context/examples what would not be "data limited" setup?

2) Page 7, last paragraph. The description of what parts were finetuned/update/adapted and when is somewhat unclear and involved. How did you come up with this setup? Could you clarify this paragraph or provide a figure to help with understand what was updated in which stage?

## Comments
- Define acornyms only once and consistently use the acornym ever since (e.g., TAIL is defined multiple times).


## Rebuttal update 20th Nov

I have read authors' answers to my questions, and I have kept my original review score.

---

> ### Author Response · Authors · 2023-11-18
> **Response to Reviewer Lua3**
>
> Thank you for your detailed and positive review! We are glad you find the paper well-written and the experiments and results comprehensive. We respond to individual concerns and questions below. We also included new experiments in the revised version of the paper to provide an even more comprehensive view of our paper.
>
> > **Only one environment?**
>
> We have added **new experiments** in Franka-Kitchen, using the same pre-trained model and following the experiment setup presented in the RoboAdapter and R3M paper. The detailed results are shown in Appendix C.7, where TAIL-LoRA performs the best, achieving an 80% success rate while RoboAdapter achieves 65%. The results indicate that our findings and claims from the extensive experiments on the large-scale LIBERO datasets can be generalized well on other small-scale domains, such as Franka-Kitchen.
>
> We would also like to emphasize that the LIBERO dataset is a large-scale, comprehensive benchmark for evaluating continual imitation learning. It comprises over 100 different tasks in various environments, including the Kitchen, Living Room, and Study Room. This dataset aims to evaluate the diverse aspects of continual learning algorithms, including spatial relationship reasoning, object recognition, and understanding of task goals. Therefore, the main results and findings of our work should be generalizable to similar robotic task setups.
>
> > **Per-task weights for TAIL is unfair comparison?**
>
> We clarify that we compare against Frozen Pre-trained Features (FPF) which also learns a new set of per-task weights. TAIL greatly outperforms this baseline in all adaptation environments (Fig 5), indicating that TAIL’s adapter integration is important for forward transfer even against baselines that learn per-task suite new weights.
>
> To further illustrate this, we also add a **new experiment** comparing against PackNet [1], which prunes weights before re-training the pruned params (as new params) for new task suites. Results demonstrate that TAIL outperforms this baseline by up to **3x** in specific task suites. See the updated Fig 5 for further details.
>
> [1]: PackNet: Adding Multiple Tasks to a Single Network by Iterative Pruning. Arun Mallya and Svetlana Lazebnik.
>
> > **“How do you consider this a ‘limited data’ setup? Can you give context/examples what would not be ‘data limited’ setup?”**
>
> Thanks for flagging this. When we talk about "limited data," we mean that only a few demonstrations are available for each task. This is a realistic assumption, considering the expense involved in collecting demonstrations. We have reworded Section 5.2 to make this clearer.
>
> > **Page 7, last paragraph: Clarify description of what parts were finetuned/update/adapted.**
>
> Thank you for the suggestion and we apologize for the confusion. Since we only have pretrained CLIP weights for the vision and text encoders, all other parameters (including the GPT-2 model) must be trained in the pretraining stage. After that, during the adaptation stage for the subsequent task suites, we add adapters for the CLIP vision and text encoders and the temporal GPT2 decoder while keeping their transformer backbone parameters frozen. Since the fusion module and policy head are lightweight, we also tune them fully for each adaptation stage and store their parameters together with corresponding adapters per task suite.
>
> In short, the CLIP spatial and instruction encoders are frozen across all stages. The temporal GPT2 decoder weights are trained in the pretraining stage (Kitchen) but frozen in the adaptation stages. The fusion module and the policy head are fully tuned with adapters per task suite. We have revised the last paragraph of Section 5.2 to address the confusion.
>
> > **Minor comments: change names, acronyms**
>
> Thank you for these suggestions. We will address this in the final revision.
>
> Please let us know if you have any more questions!

---

> > ### Comment · Reviewer_Lua3 · 2023-11-20
> >
> > Dear authors. Thank you for your extensive replies and additional results! These are very insightful, and your answers satisfy my questions. I have decided to keep the score at original 8, as the next score is 10. While good paper, and I reckon people working on same benchmarks will find many details here useful, I do not feel it is "highlight" quality for ICLR, given the somewhat limited novelty of the method.

---

> > > ### Author Response · Authors · 2023-11-22
> > >
> > > Dear reviewer, thanks for your reply. We greatly appreciate your valuable feedback in improving our paper and your acknowledgment of our work.

---

### Official Review · Reviewer_JmE1 · 2023-10-25

**Soundness:** 3 good
**Presentation:** 2 fair
**Contribution:** 2 fair
**Rating:** 6
**Confidence:** 3

**Summary:**

The authors applied parameter-efficient fine-tuning methods, including bottleneck adapters, P-tuning, and low-rank adaptation, to the continual imitation learning task. TAIL is designed to adjust large pre-trained models for new tasks even with limited demonstration data, with GPT2 as the backbone and CLIP-based modules as encoders. It avoids issues like catastrophic forgetting by activating the corresponding adapter for that task.

**Strengths:**

* The paper rigorously tests TAIL with prevalent PEFT techniques on the imitating continual imitation learning task.

* TAIL’s ability to achieve superior post-adaptation performance using only 1% of the trainable parameters of full fine-tuning highlights the framework’s efficiency, making it a potentially valuable tool for resource-constrained settings.

**Weaknesses:**

* The framework's capability to circumvent catastrophic forgetting is primarily attributed to its unique configuration that allows for the selection of adapters tailored to individual tasks. Its efficiency largely draws from existing PEFT methodologies, which limits the novelty and distinctiveness of the presented framework.

* Drawing a comparison between TAIL and the fine-tuning baseline methods cited in the paper is unfair. This is because TAIL is designed to learn dedicated adapters for each distinct task. Thus, the claims regarding its efficacy become less compelling.

**Questions:**

* Could you show the results when setting the rank for LORA to 32, 256, 512, and 768 (the dimensionality of the embeddings and hidden states of GPT2)? The ablation studies regarding the number of parameters should be added.

* Could you provide some insights on whether combining the Lora and P-tuning can improve the performance?

---

> ### Author Response · Authors · 2023-11-18
> **Response to Reviewer JmE1**
>
> Thank you for your insightful comments! We appreciate that you find the paper performs rigorous tests and our TAIL framework could be a valuable tool for resource-constrained settings. We have addressed all your comments below. We hope you consider increasing your score after seeing our detailed responses.
>
> > **Novelty: largely tests existing PEFT methodologies**
>
> Our focus is not on proposing new adapter strategies but rather on proposing a general framework (TAIL) for efficient adaptation in a continual imitation learning setting and testing various adaptation strategies for this setting. No prior work considers this adaptation setting with large pre-trained decision-making models, and prior work like RoboAdapter [1] only considers a single adapter strategy.
>
> Our study contrasts multiple adapter strategies in this setting, including the RoboAdapter strategy, and we find great performance differences among the techniques (Fig 4), with RoboAdapter performing on average the worst among all techniques in LIBERO. Our new experiments demonstrate that RoboAdapter performs worse even in Franka Kitchen, a domain from the paper.
>
> Please see the dedicated “To summarize our contributions…” paragraph in the Response to all reviewers for further details about our contributions.
>
> [1]: Lossless Adaptation of Pretrained Vision Models for Robotic Manipulation (RoboAdapter). Sharma et al, 2023.
>
> > **Comparison between TAIL and fine-tuning baselines unfair b/c TAIL learns new adapters?**
>
> We respectfully disagree that the comparison is unfair. FPF (frozen pre-trained features), like TAIL, also learns dedicated additional parameters per task (in the output head) yet performs much worse. As another comparison, the Experience Replay (ER) method maintains a data buffer for **all previous data**, which takes much more storage space than TAIL’s adapter weights. Moreover, except FPF, all other baselines (FFT, EWC, ER) are standard techniques for continual learning that tune the full parameters for each new task. In contrast, TAIL only tunes a small portion, yielding better forward transfer performance than them. Hence, TAIL doesn’t have an unfair advantage in our continual learning setting.
>
> To further illustrate this, we have also added a **new baseline**, PackNet [1], which prunes the network and then re-trains pruned params for the new task. Results demonstrate that TAIL is superior in adaptation performance for almost all task suites, up to a factor of ****3x**.** Please see the updated Fig. 5 for more details.
>
> [1]: PackNet: Adding Multiple Tasks to a Single Network by Iterative Pruning. Arun Mallya and Svetlana Lazebnik.
>
> > **Can you show LoRA dimensionality ablations?**
>
> We agree that this is an important ablation. Thus, we have performed **new experiments** ablating LoRA ranks from 2 to 768, demonstrating success rates increase to a rank of 32 and then slightly decrease as rank increases to 768. This validates our hypothesis that fine-tuning *more* parameters on limited data can lead to overfitting. These experiments and discussions have been added to Appendix C.4 and Fig. 9.
>
> > **Can you combine LoRA and P-tuning for better performance?**
>
> Thank you for this excellent suggestion! We’ve now included a **new experiment** in which we test all combinations of LoRA, bottleneck layers, and p-tuning. The results demonstrate that all combinations that include LoRA perform similarly, while all combinations without LoRA perform far worse, indicating the necessity of LoRA. See Appendix C.6 for more info.
>
> Please let us know if you have any more questions!

---

> > ### Author Response · Authors · 2023-11-22
> > **Reminder of the response**
> >
> > We thank the reviewer for your time and effort. We have provided detailed answers to your questions, **added a new baseline based on your suggestion**, and **included other ablation experiments**. We hope that our reply addresses your concerns. Since the rebuttal window is close to the end, we would be happy to address any other concerns and questions that you may have that prevent you from increasing the score.

---

> ### Comment · Reviewer_JmE1 · 2023-11-22
>
> Thank you for the comprehensive additional experimental results, which have effectively addressed my previous concerns about the ablations. Hence, I've decided to raise my scores. However, I still believe that the comparison is not entirely fair for a couple of reasons. 1) Even though ER uses all the previous data, the lack of a MoE setting may also lead to worse performance. 2) The different number of trainable parameters remains a concern. Additionally, the ablation studies related to LoRA's rank suggest that increased parameters do not necessarily equate to improved performance.

---

> > ### Author Response · Authors · 2023-11-23
> > **Thank you for increasing the score**
> >
> > We sincerely thank the reviewer for the response and increasing their score. We make the following clarifications regarding your latest comments.
> >
> > > The comparison is not entirely fair.
> >
> > **To ensure our comparisons are fair and accurate, we closely follow the setup of each continual learning method as described in their respective literature.** For example, an Experience Replay (ER)-based method requires storing all previous data, which is not a necessity for other methods. This is also true for regularization-based methods like Elastic Weight Consolidation (EWC), which necessitate maintaining a complete copy of the previous model during adaptation, unlike other approaches. As demonstrated, each method has its specific setup and requires additional resources, making it impractical to standardize every method.
> >
> > Therefore, **the challenge of continual learning lies in how to utilize the least additional resources to achieve optimal performance**. Compared with ER and EWC, our method, TAIL, is more scalable, requires fewer computing and memory resources, and delivers superior performance. As shown in the experimental section, using ER necessitates an additional 28GB of storage space for the dataset, whereas our adapter requires only 7.8MB. EWC also consumes much more computing memory than TAIL, which significantly slows down the training but still yields worse performance. **Thus, TAIL presents a significant advantage in the continual learning context.** We have included a detailed discussion on this in Appendix B.1 to mitigate any potential confusion.
> >
> > > The rank ablation study suggests that increased parameters do not necessarily equate to improved performance.
> >
> > Correct, we observe that increasing the rank size generally enhances performance up to a certain point. Beyond this optimal threshold, further increases in rank size do not necessarily lead to higher success rates. **This phenomenon**, which we aim to highlight in our ablation study, **aligns with the findings in Section 7.2 of the original LoRA paper [1]**. There, LoRA already performs competitively with a small rank, suggesting that the update matrix ∆W per task could have a very small “intrinsic rank” [2]. In other words, **increasing the rank does not cover a more meaningful subspace, indicating that a low-rank adaptation matrix is often sufficient.**
> >
> > This observation also aligns with our findings that using full parameter fine-tuning (full-rank tuning) with limited data can yield worse results compared to TAIL, potentially due to overfitting issues, as discussed in Appendix C.1. Therefore, our TAIL framework exhibits data adaptivity, meaning that the rank size can be adjusted based on the quantity of adaptation data. In scenarios with smaller datasets, a smaller rank size should be sufficient to solve the task, and vice versa. We have added related discussions in Appendix C.4.
> >
> > Thank you again for your valuable comments, which have helped to improve our work. We hope our responses fully address your concerns.
> >
> > [1] Hu, Edward J., et al. "LoRA: Low-Rank Adaptation of Large Language Models." International Conference on Learning Representations. 2021.
> >
> > [2] Aghajanyan, Armen, Luke Zettlemoyer, and Sonal Gupta. "Intrinsic dimensionality explains the effectiveness of language model fine-tuning." arXiv preprint arXiv:2012.13255 (2020).

---

### Official Review · Reviewer_FCxQ · 2023-10-28

**Soundness:** 2 fair
**Presentation:** 3 good
**Contribution:** 2 fair
**Rating:** 5
**Confidence:** 4

**Summary:**

The paper considers the adaptation problem in continual imitation learning and proposes task-specific adapters when fine-tuning the imitation policy to new tasks. Specifically, the task-specific adapter is an add-on to the imitation policy and can be updated for each task without changing the pretrained weights in the backbone policy. As shown by the paper, this adapter can be implemented by three ways: parallel, i.e., Low-Rank Adaptation (LoRA) in this paper, sequential, i.e., Bottleneck Adapter, & prefix token, i.e., prompt-tuning. The paper evaluates the proposed adapter in LIBERO robotic manipulation continual learning benchmark and shows that LoRA adapter performs the best across all tasks.

**Strengths:**

### Clarity
The structure and presentation of the paper is of very good quality. The flow of the paper is excellent with a good and appropriate structure. The underlying motivation of studying the task-specific adapters is well justified.

**Weaknesses:**

### Quality & Contribution
One of the main weaknesses of this paper is the lack of comprehensive empirical analysis over multiple different benchmarks, especially those used by previous baseline methods. Particularly, **the conclusions and empirical insights were drawn only from a single dataset, with no results from any other public datasets**. As the main contribution, this paper shows that the LoRA-type integration of the adapter in continual imitation learning performs the best. The paper thus concludes that they “are contrary to many results from the vision and language model literature which show that full fine-tuning works better”. However, these results are drawn based on LIBERO robotic manipulation continual learning benchmark ONLY. They have not been verified by any other datasets. It is thus unclear if and how the results and conclusions can be generalized to other robot control tasks, e.g., Franka-Kitchen.

Furthermore, it is unclear if the baseline methods have been well tuned for the LIBERO benchmark. For example, the RoboAdapter method may need to tune the adapter locations and different pre-trained representations for its optimal performance. The paper, however, seems to take the default configs, even though those configs were originally proposed for other robot tasks. To have a fair comparison with RoboAdapter, the paper should consider some of datasets used by RoboAdapter paper, i.e., Metaworld, Franka-Kitchen, and RGB-Stacking task suites, replicate the reported performance of RoboAdapter and then demonstrates on at least one of those datasets that the TAIL performs better than RoboAdapter.

### Originality & Significance
The originality of this paper can be limited and incremental.

**In terms of technique**, the adapter idea has been well studied by the RoboAdapter paper, which introduces the adapter layer (Sequential Integration in this paper) to imitation learning and considers its application in robot controls. Though the paper proposes two other adapter mechanisms, LoRA & Prefix prompt tuning, both of them are taken directly from other papers, without any substantial modifications or creative combinations to the continual imitation learning or robot control tasks.

**In terms of the empirical results**, they may not offer any in-depth understanding of using adapters in continual imitation learning and the significance can be limited. The paper mentions that TAIL with LoRA “avoiding catastrophic forgetting and preserving adaptation plasticity”. Avoiding catastrophic forgetting can be evident since the adapter weights are task-specific and not shared across different tasks. But the empirical results shed little light on the adaptation plasticity. It is unclear if and how the adaptation plasticity is preserved given that the empirical results focus only on the success rate, as in Figure 4.

**Questions:**

1. In appendix, “all methods share similar amount of parameters”. I’m not quite sure how to interpret this. Why all methods need to have a similar number of parameters? Even though the adapter integration mechanisms are different? Could this constraint on the number of parameters be biased towards to LoRA adapter? (since “filtering often requires a larger bottleneck size compared to that of LoRA, leading to more parameters”)

2. In the experiment Table 1, “The BWT for TAIL methods are all 0 (no catastrophic forgetting)”. Is it self-evident? since the TAIL methods produce an ensemble of models (fixed pretrained weights + task-specific adapter weights), each of which aims to solve a dedicated task and is updated independently to each other. Then why do we need to use BWT as a performance indicator.

3. The colours for Prefix Token & Frozen cannot be easily distinguished in Figure 2.

---

> ### Author Response · Authors · 2023-11-18
> **Response to Reviewer FCxQ (Part 1/2)**
>
> Thank you for the detailed review. We are happy you found the presentation very good and our motivation excellent. We have addressed all your comments below. We hope you consider increasing your score after seeing our detailed responses and new experiments.
>
> > **Draw conclusions from more environments, e.g, Franka Kitchen from RoboAdapter?**
>
> Thank you for the suggestion; we have now added **new experiments** in Franka-Kitchen, using the same pre-trained model and following the experiment setup presented in the RoboAdapter and R3M [1] paper. The detailed results are shown in Appendix C.7, where our TAIL-LoRA performs the best, achieving an 80% success rate while RoboAdapter achieves 65%. The results indicate that our findings and claims from the extensive experiments on the large-scale LIBERO datasets can be generalized well on other small-scale domains, such as Franka-Kitchen.
>
> We would also like to emphasize that the LIBERO dataset is a large-scale, comprehensive benchmark for evaluating continual imitation learning. It comprises over 100 different tasks in various environments, including the Kitchen, Living Room, and Study Room. This dataset aims to evaluate the diverse aspects of continual learning algorithms, including spatial relationship reasoning, object recognition, and understanding of task goals. Therefore, the main results and findings of our work should be generalizable to similar robotic task setups.
>
> We hope this addresses your quality & contribution concerns.
>
> [1] Nair, Suraj, et al. "R3m: A universal visual representation for robot manipulation." *arXiv preprint arXiv:2203.12601* (2022).
>
> > **Originality: adapters studied in other works?**
>
> Our focus is not on proposing new adapter strategies but on proposing a general framework (TAIL) for efficient adaptation in a continual imitation learning setting and testing various adaptation strategies for this setting. No prior work considers this adaptation setting with large pre-trained decision-making models, and prior work like RoboAdapter [1] only considers a *single* adapter strategy.
>
> Our study contrasts multiple adapter strategies in this setting, including the RoboAdapter strategy, and we find great performance differences among the techniques (Fig 4), with RoboAdapter performing on average the worst among all techniques in LIBERO. Our new experiments demonstrate that RoboAdapter performs worse even in Franka Kitchen, a domain from the paper.
>
> Please see the dedicated “To summarize our contributions…” paragraph in the Response to all reviewers for further details about our contributions.
>
> [1]: Lossless Adaptation of Pretrained Vision Models for Robotic Manipulation (RoboAdapter). Sharma et al, 2023.

---

> ### Author Response · Authors · 2023-11-18
> **Response to Reviewer FCxQ (Part 2/2)**
>
> > **Significance: results don’t focus on adaptation plasticity?**
>
> We clarify that we **do** have adaptation plasticity experiments in Table 2 in the main paper and Appendix Table 6. Upon re-visiting previously seen task suites, full fine-tuning suffers large drops in performance despite re-training on the same data (Table 2)
>
> Meanwhile, because TAIL allows us to simply store **~7.8MB** of weights per task suite for later reuse, we can reuse the weights when returning to previously seen tasks. As a comparison, storing the entire Kitchen datasets (for the ER method) and the entire model per task suite takes **28GB* and *660MB*, respectively. Furthermore, adding TAIL w/ LoRA **after full fine-tuning** is worse than LoRA without full fine-tuning on all tasks (Table 6), indicating that full fine-tuning leads to adaptation plasticity loss.
>
> We have clarified in Section 5.3 and Appendix C of the revised paper that these experiments are studying **adaptation plasticity** specifically. We hope we have addressed both of your main concerns about originality & significance!
>
> > **Why do all methods share similar # params? Does this favor LoRA?**
>
> We kept the parameter count the same to study how effective adaptation techniques were under the same memory constraint. However, per your request, we have performed a **new experiment** ablating the bottleneck layer size for the sequential Bottleneck Adapter method, demonstrating that LoRA still outperforms in rank scaling — achieving nearly double the performance at identical rank in some instances. Notably, the sequential Bottleneck Adapters with full rank size, i.e., 768, still under-perform LoRA with rank size 8, showing advantages of the parallel integration style in the continual learning setup. We have added this to Appendix C.4 of the revised paper.
>
> > **Why study Backwards Transfer (BWT) in TAIL methods? Isn’t it self-evident?**
>
> BWT is a relevant and important metric for all non-TAIL methods we compare; it helps analyze catastrophic forgetting among our baselines. In our setting, BWT transfer is poor even among methods made explicitly to resolve this issue (Experience Replay, Elastic Weight Consolidation), indicating the need for TAIL (see Table 1).
>
> > **Fig 2: colours hard to distinguish**
>
> Thank you, we have updated the colors.
>
> Please let us know if you have any more questions!

---

> > ### Comment · Reviewer_FCxQ · 2023-11-21
> > **Post-rebuttal feedback**
> >
> > Thank you for the response. Since this paper focuses on the empirical analysis of the task-specific adapters in continual learning, a bit more comprehensive analysis on many different benchmarks would be expected (which would strengthen the conclusion as well). Also, a detailed description of the baseline setups may be required. I appreciate the value of this work and also the authors' extra efforts in providing the empirical results in FrankaKitchen. But I think the work in its current version may be below the acceptance threshold.

---

> > > ### Author Response · Authors · 2023-11-22
> > > **Further response to reviewer FCxQ**
> > >
> > > We thank the reviewer for acknowledging the value of our work. However, we are a bit confused about your concerns regarding our paper, as we believe we have addressed all of the issues raised initially. We hope the reviewer can share any remaining concerns they have so we can address them accordingly.
> > >
> > > In particular, we have conducted comprehensive experiments over **40 pretraining tasks** and **55 adaptation tasks** across **two different environments**, with detailed ablation studies for the key components in our method.
> > >
> > > We emphasize that the LIBERO dataset is a large-scale, comprehensive benchmark for evaluating continual imitation learning. It comprises tasks in various environments, including the Kitchen, Living Room, and Study Room. This dataset aims to evaluate the diverse aspects of continual learning algorithms, including spatial relationship reasoning, object recognition, and understanding of task goals. Therefore, the main results and findings of our work should be generalizable to similar robotic task setups.
> > >
> > > Our **additional experiments** in the Franka Kitchen environment as requested by the reviewer **further validate the effectiveness of our approach and strengthen the claims in our paper**.
> > >
> > > Regarding the baseline setups, we have a detailed description of each baseline approach in Appendix B.1, including their method implementations, corresponding analysis, and insights.  We would like to remind the reviewer to check the updated Appendix C (highlighted in red) for the additional experiments.

---

### Official Review · Reviewer_86yf · 2023-10-29

**Soundness:** 3 good
**Presentation:** 3 good
**Contribution:** 3 good
**Rating:** 6
**Confidence:** 3

**Summary:**

The paper considers the continual imitation learning setting where a stream of tasks arrive one at a time. Existing methods are susceptible to catastrophic forgetting or loss of model plasticity as the number of tasks increase. Consequently, the paper proposes task-specific adapters for imitation learning (TAIL), a method that uses additional per-task tuneable parameters and synthesizes them to the pretrained base model's parameters. The paper demonstrates through experiments that TAIL introduces little computational overhead and mitigates catastrophic forgetting and model plasticity problems.

**Strengths:**

- The paper is easy to follow generally.
- The method is very intuitive and simple---the fact that the overhead is low for a new task while mitigating catastrophic forgetting (e.g. better than experience replay) is appealing.
- The empirical analysis demonstrates that using TAIL with low-rank adaptation (LoRA) perform better than existing baselines convincingly in LIBERO, interestingly that it performs that well with rank $r = 8$.

**Weaknesses:**

**Comments**
- It appears that the number of tasks can easily explode over time since each task instruction (and initial-state distribution) corresponds to a new task. A natural extension will be to mitigate the amount of adapters based on the similarity of task instructions, as hinted in appendix.
- Since the problem setting indicates that a task definition is not w.r.t. the state-action space, it will be more convincing if there are experiments conducted on cross-embodiment (e.g. different arms or same arm with different inertial properties.)
- The evaluation metric is unclear---in particular under BWT, the equation seems to be different from what is described---should the *best FWT model* be the best $F_i$ for task $i$ after seeing $k$ tasks?
- For the 10 validation episodes, I believe it has been shown multiple times that validation error does not necessarily correspond to success rates [1, 2, 3]---a model may achieve high validation error while still achieving high success rate.
- Regarding success rates, what are the baseline performances? That is, what is the success rate of the expert demonstrations? Do we expect the policy that trains purely using the task-specific data to perform better than the models in Figures 5 and 6? How would a random policy perform?

**References**
[1]: Hussenot, L., Andrychowicz, M., Vincent, D., Dadashi, R., Raichuk, A., Ramos, S., ... & Pietquin, O. (2021, July). Hyperparameter selection for imitation learning. In International Conference on Machine Learning (pp. 4511-4522). PMLR.
[2]: Mandlekar, A., Xu, D., Wong, J., Nasiriany, S., Wang, C., Kulkarni, R., ... & Martín-Martín, R. (2021). What matters in learning from offline human demonstrations for robot manipulation. arXiv preprint arXiv:2108.03298.
[3]: Ablett, T., Chan, B., & Kelly, J. (2023). Learning from Guided Play: Improving Exploration for Adversarial Imitation Learning with Simple Auxiliary Tasks. IEEE Robotics and Automation Letters.

**Questions:**

- Under subsection **Training, Adaptation, and Evaluation**, what exactly is a validation scene?
- Does ER retrain parameters from scratch, or does it continue training from the current set of parameters? Clearly if it is the latter the model will experience plasticity loss similar to the reinforcement learning setting [1].
- It appears that some tasks are repeated based on appendix. I am wondering if there is any result regarding the per-task success rate. Do we understand whether the parameters are well-adapted to specific tasks, or are they similar in performance?

**Possible typos**
- On page 2, subsection PEFT, line 5: "It is" instead of "it's".

**References**
[1]: Nikishin, E., Schwarzer, M., D’Oro, P., Bacon, P. L., & Courville, A. (2022, June). The primacy bias in deep reinforcement learning. In International conference on machine learning (pp. 16828-16847). PMLR.

---

> ### Author Response · Authors · 2023-11-18
> **Response to Reviewer 86yf**
>
> Thank you for the insightful review. We appreciate that you find the paper easy to follow, intuitive, and that TAIL performs well. We have addressed all your comments below. We hope that you consider increasing your score after seeing our detailed responses and new experiments.
>
> > **Cross-embodiment tasks?**
>
> This is a great suggestion. Following your suggestion, we have added **new experiments** to use the model pre-trained on LIBERO data to adapt to the Franka kitchen tasks. The detailed results are shown in Appendix C.7, where TAIL-LoRA performs the best, achieving an 80% average success rate while the best baseline achieves 67%. The results indicate that our findings and claims from the extensive experiments on the large-scale LIBERO datasets can be generalized well on other small-scale domains, such as Franka-Kitchen.
>
> > **“Do we expect the policy that trains purely using the task-specific data to perform better than the models in Figures 5 and 6? How would a random policy perform?”**
>
> Thank you for the suggestion regarding the task-specific agent and random agents. First, using a random policy without any fine-tuning always results in a 0 success rate, since our tasks are relatively long and generally require 200-400 steps. However, per request, we added **new ablation studies** for the task-specific agent in Appendix C.5 of the revised paper.
>
> In summary, if we only train the model with corresponding task data, it does perform better than sequential full fine-tuning in the later stages. However, our TAIL-LoRA still outperforms this task-specific model. In addition, maintaining an entire copy of the model for each adaptation task consumes much more storage space than our approach, and as shown in our results in Table 2 and Appendix C.3, training a large model on a narrow domain with limited data could reduce the model’s plasticity in adapting to new tasks. Therefore, these results further strengthen the advantage of our approach in terms of both resource efficiency and performance.
>
> > **Tasks explode over time b/c new adapter for each task?**
>
> We use a single adapter for each task *suite,* which is of at least 8 tasks, corresponding to just 5 adapters for the 5 task suites (totaling 40 different tasks) in LIBERO in Fig. 5. We have clarified this in Section 5.2 of the revised paper. While we agree with your suggestion about using adapters based on the similarity of task instructions as it would be an exciting direction, it is beyond the scope of this work and we leave it for future research direction.
>
> > **Unclear evaluation metric**
>
> Thanks for flagging this; there is indeed a typo for the BWT metric calculation. The BWT metric for the k-th task is computed by:
> $B_k = \frac{1}{k-1} \sum_{i=1}^{k-1} (S_{i} - F_{i})$,
> where $S_{i}$ values are from the best FWT model at task k, and $F_{i}$ values are from the best FWT model in the corresponding i-th task. We have revised this in Section 5.2.
>
> > **Validation error doesn’t correspond to success rate?**
>
> To clarify, we do not use the validation error to perform model selection. Success rates are simply computed either at the end of adaptation fine-tuning (Table 1) or displayed as a curve over training epochs (Fig 5) (see “training, adaptation, and evaluation” in Section 5.2). As mentioned by your referred paper Appendix G [1], validation loss is in fact a poor measure of policy performance. The purpose of displaying the validation errors is to show that full fine-tuning has a higher risk to overfit to the small data.
>
> [1]: Mandlekar, A., Xu, D., Wong, J., Nasiriany, S., Wang, C., Kulkarni, R., ... & Martín-Martín, R. (2021). What matters in learning from offline human demonstrations for robot manipulation. arXiv preprint arXiv:2108.03298.
>
> > **“Regarding success rates, what are the baseline performances? That is, what is the success rate of the expert demonstrations?”**
>
> The success rate of expert demonstrations is 100%; we have clarified this in Section 5.2.
>
> > **Some tasks are repeated in evaluation?**
>
> Although specific tasks may share similar descriptions, they are not identical due to environment variations, including goals, object arrangements, or object layouts (see Fig 3 for concrete examples). Consequently, these tasks are distinct. We have updated Appendix Section D with this discussion.
>
> > **What’s a validation scene?**
>
> Each task comes with 50 demonstrations with different scene setups. We use 40 for training and 10 for validation. The validation scenes are the scenes from which the validation trajectories were created. We have clarified this in Section 5.2 of the revised paper.
>
> > **Does ER retrain from scratch or continue fine-tuning?**
>
> Experience replay (ER) continues fine-tuning—we do not retrain from the original pre-trained model as it would require storing the entire model per task suite and is extremely storage inefficient for the continual learning setting.
>
> Please let us know if you have any more questions.

---

> > ### Comment · Reviewer_86yf · 2023-11-21
> >
> > Thank you for addressing all the points. I agree with reviewer Lua3 that this will be very useful work to benchmark against.
> > However, my understanding is also that this is under supervised imitation learning setting, thus there remains a question about these methods under (inverse) reinforcement learning. With this point along with the discussed limitations, I will keep my score.

---

> > > ### Author Response · Authors · 2023-11-22
> > > **Response to Reviewer 86yf**
> > >
> > > Dear reviewer, we deeply appreciate your thoughtful feedback. There have been notable recent work utilizing large, pre-trained decision-making models for imitation learning, such as [RT-1](https://arxiv.org/abs/2212.06817), [RT-2](https://arxiv.org/abs/2307.15818), [RoboCat](https://arxiv.org/abs/2306.11706), and [VIMA](https://arxiv.org/abs/2210.03094). In light of these advances, we believe TAIL makes a valuable contribution by studying efficient adaptation methods for these models under data and resource constraints. Such capabilities are increasingly relevant as these imitation learning models see wider adoption.
> > >
> > > We do agree that studying efficient adaptation under (inverse) RL is also an important direction, and we hope that if published,  TAIL can serve as a framework future work can build on to study that setting.
> > >
> > > We appreciate the opportunity to discuss this further, please let us know if there are any remaining unaddressed points!

---

> > > > ### Comment · Reviewer_86yf · 2023-11-22
> > > >
> > > > Thank you for the discussions, I believe the paper is fairly clear based on my current understanding.

---

### Official Review · Reviewer_q7Ls · 2023-10-30

**Soundness:** 3 good
**Presentation:** 3 good
**Contribution:** 3 good
**Rating:** 6
**Confidence:** 4

**Summary:**

This paper introduces a new architecture named TAIL for imitation learning, which effectively adapts large pretrained models to new tasks with limited demonstration data. The main contribution of this method is to effectively incorporate lightweight adapter modules into pretrained models using various integration techniques, including parallel integration (with LoRA weights), sequential integration (Bottleneck Adapter), and prefix token integration.

The paper provides a comprehensive comparison of these adaptation techniques on the LIBERO robotic manipulation continual learning benchmark. The results indicate that TAIL, particularly when utilizing LoRA integration, outperforms the compared methods in terms of both forward and backward transfer.

Overall, while this paper may not introduce strong technical novelty, it offers a good empirical study of lightweight continual learning techniques for imitation learning of robot control tasks.

**Strengths:**

1. The paper is well-written and easy to follow.
2. The paper includes sufficient experiments on the LIBERO benchmark, which can effectively demonstrate the advantages of employing TAIL in conjunction with the LoRA integration technique.
3. The proposed TAIL-based methods greatly outperform the conventional fine-tuning method in both forward transfer results and adaptation efficiency.

**Weaknesses:**

1. My primary concern is the technical novelty of this paper, although I acknowledge the significance of benchmarking and conducting an extensive empirical investigation of existing parameter-efficient fine-tuning techniques. The proposed TAIL framework, which incorporates lightweight adapters and task-specific heads, shares similarities with existing methods in continual learning and multi-task learning, as demonstrated in the paper from Rebuffi et al. (2017) titled 'Efficient Parametrization of Multi-Domain Deep Neural Networks.' Although TAIL introduces a new multi-modal, Transformer-based architecture, the core idea bears a strong resemblance to prior work.
2. In Figure 4, the authors primarily compare various design choices within the TAIL framework when presenting forward adaptation results. It would be valuable if the authors could extend this analysis to include a comparison of TAIL's best performance with the state of the art in lifelong imitation learning. Furthermore, in Table 1, the comparison of TAIL's forward transfer performance with previous continual learning methods may not be fair enough. This is because EWC and ER primarily address catastrophic forgetting and may not contain task-specific model parameters.
3. In Figure 5, a comparison between TAIL and EWC/ER may not be entirely equitable, considering that these prior methods do not assume known task IDs or retain task-specific model parameters. I would like to suggest the authors include a comparison with multi-task learning methods for a more comprehensive evaluation.

**Questions:**

1. Is a separate adapter $\omega_k$ trained for each task suite? My understanding is that when re-evaluating previous task suites $j$, the previously trained adapter $\omega_j$ is reloaded instead of testing the previous task on the adapter $\omega_k$ trained for the current task $k$.
2. In Figure 4, it would be insightful to determine whether combining all integration techniques, including LoRA, Bottleneck Adapter, and Prefix Token, would yield further performance improvements.

---

> ### Author Response · Authors · 2023-11-18
> **Response to Reviewer q7Ls**
>
> Thank you for your detailed review. We appreciate that you found our paper well-written and the experiments sound! We have addressed all your comments below. We hope you consider increasing your score after seeing our detailed responses and new experiments.
>
> > **What is the technical contribution?**
>
> Please see our answer to this above in the **Response to all reviewers** that addressed this question: “To summarize our contributions…”
>
> > **“The proposed TAIL framework, which incorporates lightweight adapters and task-specific heads, shares similarities with existing methods in continual learning and multi-task learning, as demonstrated in the paper from Rebuffi et al. (2017)”**
>
> Thank you for this pointer. While both papers have similar motivations, there are major differences. In particular, Rebuffi et al does not (1) study continual imitation learning, (2) investigate decision-making settings, (3) or investigate transformer-based models.
>
> (1) and (2) are important for deploying decision-making agents in realistic settings (e.g., robots) and (3) is important given the impact that transformers have had on large-scale pre-training.
>
> We’ve now discussed Rebuffi et al. in our updated related works section.
>
> > **Compare TAIL against SOTA in continual imitation learning?**
>
> That is a good suggestion. We have now added a comparison against PackNet [1], a SOTA method as shown in the original LIBERO paper, which prunes parameters to be then re-learned for every new task. The detailed results are updated in Fig. 5 of the revised paper. TAIL with LoRA outperforms PackNet in forward imitation in every task suite, for example, **2 times** in the Goal task suite and **3 times** in the Living Room task suite. Furthermore, PackNet still suffers from severe catastrophic forgetting problems after long continual adaptation stages while TAIL does not.
>
> [1]: PackNet: Adding Multiple Tasks to a Single Network by Iterative Pruning. Arun Mallya and Svetlana Lazebnik.
>
> > **“In Figure 5, a comparison between TAIL and EWC/ER may not be entirely equitable, considering that these prior methods do not assume known task IDs or retain task-specific model parameters”**
>
> We clarify that all methods can access task ID: the language descriptions as shown in Fig 1. Thus, task-specific model parameters are also contained in EWC/ER in the language encoder and the entire base model is fine-tuned in EWC/ER so that they can fairly perform forward transfer. We have clarified this in Section 5.2 of the revised paper.
>
> > **Try combining LoRA, bottleneck, and p-tuning?**
>
> Thank you for this excellent suggestion! We’ve now included **new experiments** in Appendix C.6  of where we test all combinations of LoRA, bottleneck layers, and p-tuning. All combinations including LoRA perform similarly (75% success rate) and significantly better than other combinations (41-65%), indicating that LoRA is necessary for good adaptation performance.
>
> > **Separate adapter for each task suite?**
>
> Yes, we include a separate adapter for each task suite. We have clarified this in Section 5.2.
>
> Please let us know if you have any more questions.

---

> > ### Author Response · Authors · 2023-11-22
> > **Reminder for the response**
> >
> > We thank the reviewer for your time and effort. We have provided detailed answers to your questions, **added a new baseline based on your suggestion**, and **included other ablation experiments**. We hope that our reply addresses your concerns. Since the rebuttal window is close to the end, we would be happy to address any other concerns and questions that you may have that prevent you from increasing the score.

---

### Author Response · Authors · 2023-11-18
**General Response to All Reviewers**

We thank all of the reviewers for their constructive comments and thorough reviews! We are glad to see that reviewers believe our paper to be well-written (q7Ls, 86yf, FCxQ, Lua3), the experiments thorough (q7Ls, 86yf, JmE1, Lua3), and the motivation justified (FCxQ). We have addressed each reviewer’s concern individually.

Following reviewer suggestions,  we have included the following new results in the revised paper:

1. **Per FCxQ and Lua3’s suggestions, we have added a new environment** to the paper: Franka Kitchen. We utilize the same tasks and setup from RoboAdapter [1], allowing us to compare with RoboAdapter’s adaptation strategy directly. Our method achieves the highest post-adaptation success rate of 80% vs the best baseline at 67%. Including Franka-Kitchen, we now evaluate adaptation to **55 total tasks** in our paper.
2. Moreover, we have added PackNet [1] per request of **q7Ls, JmE1, and Lua3 as a new baseline,** which is well-suited for learning multiple tasks. We pre-train PackNet with the same model architecture and adapt it to our entire LIBERO task suite in the same continual learning setup. Our method consistently outperforms PackNet at every adaptation stage, working up to **3x** better than PackNet on certain task suites.
3. Finally, we have added **2** **additional ablation studies** comparing (1) success rate vs. adapter rank size per request of **FCxQ** and **JmE1**, demonstrating that TAIL w/ LoRA is better at every adapter rank than Bottleneck Adapters; (2) testing all combinations of TAIL adapters per request of **q7Ls** and **JmE1,** demonstrating that all combinations including LoRA are better than any combination without LoRA, further validating our original conclusion of TAIL w/ LoRA being the best-performing instantiation of TAIL.

These new results further demonstrate the effectiveness and wide applicability of our method.

**To summarize our contributions:**

We propose the TAIL framework for efficient adaptation techniques of large, pre-trained decision-making models in a continual learning setting. This setting is important to study separately, as it represents a realistic scenario in which a pre-trained agent (e.g., a factory-pretrained robot) needs to learn new tasks without forgetting prior ones or losing adaptation plasticity (e.g., the robot is deployed to your home).

Efficient adaptation has been studied in NLP but is under-explored in decision-making settings. Yet prior decision-making works [2, 3] only test single-task adaptation. Our continual learning setting brings out additional problems (catastrophic forgetting, loss of plasticity) to mitigate.

We integrate techniques like LoRA, Bottleneck, and P-tuning into TAIL and demonstrate that TAIL with LoRA works best in this setting and outperforms prior single-task adaptation work [2].

Our comprehensive experiments, now over **55** total adaptation tasks across different environments, demonstrate that TAIL with LoRA significantly outperforms all other baselines in adaptation. Meanwhile, we demonstrate that because adapter parameters are only about ~8MB per task suite, it is extremely simple and efficient to use TAIL to store parameters to avoid catastrophic forgetting while maintaining adaptation plasticity.

We have clarified our contributions in the updated abstract and related works sections.

[1]: PackNet: Adding Multiple Tasks to a Single Network by Iterative Pruning. Arun Mallya and Svetlana Lazebnik.

[2]: Lossless Adaptation of Pretrained Vision Models for Robotic Manipulation (RoboAdapter). Sharma et al, 2023.

[3]: Efficient parametrization of multi-domain deep neural networks. Rebuffi et al, 2018.

---

### Meta-Review · Area_Chair_imoz · 2023-12-07

**Metareview:**

This paper explores the use of parameter efficient fine tuning for continual adaptation of imitation learning policies - an important challenge (and opportunity) given current trends towards large foundation models in decision making. After initial feedback from the reviewers, the authors added additional experiments to contribute a sufficiently extensive empirical evaluation across a large number of tasks in 2 environments with comparison to suitable recent work. This work provides further evidence of how to utilize PEFT to update policies.

**Justification For Why Not Higher Score:**

+ Iterates on established methods. Contributing further evidence of interest to a specific sub-group of ICLR attendees.

**Justification For Why Not Lower Score:**

+ Addresses an important challenge (and opportunity) given current trends towards large foundation models in decision making
+ Sufficiently extensive empirical evaluation across a large number of tasks in 2 environments with comparison to suitable recent work

---

### Decision · Program_Chairs · 2024-01-16

Accept (poster)